# Magnetic Janus origami robot for cross-scale droplet omni-manipulation

Shaojun Jiang[1,2], Bo Li[3], Jun Zhao[4], Dong Wu [1] ✉, Yiyuan Zhang[2], Zhipeng Zhao[2], Yiyuan Zhang[1], Hao Yu[3], Kexiang Shao[1], Cong Zhang[1], Rui Li[1], Chao Chen[1], Zuojun Shen [5], Jie Hu[5], Bin Dong[1], Ling Zhu[4], Jiawen Li [1], Liqiu Wang [6] ✉, Jiaru Chu[1] & Yanlei Hu[1] ✉

The versatile manipulation of cross-scale droplets is essential in many fields. Magnetic excitation is widely used for droplet manipulation due to its distinguishing merits. However, facile magnetic actuation strategies are still lacked to realize versatile multiscale droplet manipulation. Here, a type of magnetically actuated Janus origami robot is readily fabricated for versatile cross-scale droplet manipulation including three-dimensional transport, merging, splitting, dispensing and release of daughter droplets, stirring and remote heating. The robot allows untethered droplet manipulation from ~3.2 nL to ~51.14 µL. It enables splitting of droplet, precise dispensing (minimum of ~3.2 nL) and release (minimum of ~30.2 nL) of daughter droplets. The combination of magnetically controlled rotation and photothermal properties further endows the robot with the ability to stir and heat droplets remotely. Finally, the application of the robot in polymerase chain reaction (PCR) is explored. The extraction and purification of nucleic acids can be successfully achieved.

Versatile manipulation of micro and nano droplets is essential in many fields. From the perspective of practical application, an efficient droplet manipulation technique should be multi-functionally integrated, that is, it can realize the transport, merging, mixing, splitting, dispensing, and even heating of the droplets[1-3]. The highly integrated droplet manipulation strategy can simplify the technological process and facilitating widespread adoption in routine laboratories[3,4]. Furthermore, the ideal droplet manipulation strategy should also be cross-scale applicable[1]. For example, it has high demands on the manipulation of nL-scale droplet in the field of high-precision analysis, while in other applications, such as microchemical plant, it requires manipulation of µL-scale droplets[5-7]. So far, various external excitations have been used to manipulate droplets such as

optical, electrical, acoustic and magnetic fields[8]. Among them, the magnetic manipulation is a fascinating way mainly owing to its unique advantages, such as not requiring complicated circuits and being independent of environmental transmittance and substrate surface charge[9,10].

To date, several magnetic actuation droplet manipulation strategies have been developed and they can be generally classified into two main categories: (I) adding magnetic additives into droplets[11-16] and (II) adding magnetically responsive materials into the elastomeric substrates[17-21]. In the first category, the magnetic additives such as hydrophobic magnetic powder[16]/hydrophilic magnetic particles[11]/steel beads[15]/printed solid frame[12]/ferrofluid[13,14] are added to the interior or surface of the droplet so that the droplet can be directly manipulated

[1]CAS Key Laboratory of Mechanical Behavior and Design of Materials, Department of Precision Machinery and Precision Instrumentation, University of Science and Technology of China, Hefei, Anhui 230027, China. [2]Department of Mechanical Engineering, The University of Hong Kong, Hong Kong, China. [3]CAS Key Laboratory of Mechanical Behavior and Design of Materials, Department of Modern Mechanics, University of Science and Technology of China, Hefei 230027, China. [4]Center of Engineering Technology Research for Biomedical Optical Instrument, Hefei Institutes of Physical Science, Chinese Academy of Sciences, Hefei, Anhui 230031, China. [5]Department of Clinical Laboratory, The First Affiliated Hospital of USTC, Division of Life Sciences and Medicine, University of Science and Technology of China, Hefei, Anhui 230001, China. [6]Department of Mechanical Engineering, The Hong Kong Polytechnic University, Hong Kong, China. ✉e-mail: dongwu@ustc.edu.cn; liqiu.wang@polyu.edu.hk; huyl@ustc.edu.cn

by the magnetic field for transport, merging, mixing and dispensing. However, the magnetic additives cause undesired contamination and additional purification steps are required to remove the additives to obtain pure droplet[9]. Moreover, some additives such as ferrofluid may be incompatible with biological applications, which hinders the application of droplet manipulation in biomedical fields[22]. In the latter category, the magnetic additive is added to the elastomeric substrates to prepare the magneto-responsive surfaces with microcolumns[17], microdimple[18], microcilia[19], microplate[20], or microtube[21] structures. Droplet can be manipulated by the deformation of the structures. Contamination of droplet by additives, and additional purification steps can be avoided. However, due to the fixed substrate and the simple bending deformation properties of the structure, only ordinary droplet manipulations such as propulsion, mixing, capture, and release can be achieved, while droplet splitting and dispensing have not been realized yet[17–21]. In summary, although both categories of magnetic actuation droplet manipulation strategies mentioned above enable basic droplet manipulations, such as transport, merging and mixing, there are still some limitations (Supplementary Tables 1, 2): (a) Droplet dispensing/splitting remains a challenge, which need to rely on structural morphology or surface energy traps[11,14]. (b) Incapable of on-demand droplet release after dispensing. The dispensed droplets remain entangled with the magnetic additives[11,12,14,15] or are pinned on the surface energy traps[11], so that they cannot be released freely as the pure droplets (i.e., without magnetic additives). Furthermore, due to the influence of manipulation strategy and preparation accuracy, the magnetic actuation droplet manipulation strategies mainly handle μL-scale droplets, and are difficult to be applied to nL-scale droplets[9,10,23,24]. Therefore, the diverse manipulation of droplet that highly integrate basic magnetic actuation droplet manipulations and further extend functions such as daughter droplet dispensing, on-demand release, and other functions is still an urgent need. On top of this, extending diverse droplet manipulation to cross-scale droplets from nL-scale to μL-scale remains an ultimate challenge.

Here, we present a facile and versatile strategy to achieve omni-manipulation of micro- and nanoliter droplets by the magnetic actuated Janus origami robot (JO-robot). Compared with the existing magnetic actuation droplet manipulation methods, the combination of origami deformation ability and Janus wetting characteristic endows JO-robot with unique droplet manipulation abilities. JO-robot eliminates the need for purification and extends droplet manipulation capabilities, it can realize: (a) droplet omni-manipulation, including three-dimensional transport, merging, splitting, pure daughter droplet dispensing and release, stirring, and photothermal stirring, (b) cross-scale droplet (from several nanoliters to tens of microliters) manipulation. The high stability of this strategy imparts the JO-robot the cross-scale droplet manipulation capabilities. Versatile droplet manipulation is possible for droplets ranging from ~3.2 nL to ~51.14 μL. The JO-robot can tumble under the magnetic field and spontaneously wrap droplet for directional transportation. From the functional viewpoint, the JO-robot can be seen as the smart device that can be freely loaded and removed from the droplet, and assist droplet to complete a series of tasks with its Janus characteristic and flexible deformability. It can dispense daughter droplets with the minimum volume of ~3.2 nL. The pure daughter droplet can also be released on-demand by JO-robot (minimum volume of ~30.2 nL) without any assistance of surface topography or surface energy traps. The JO-robot can rotate to realize controllable droplets stirring. The photothermal property of the JO-robot is also explored for remote heating while stirring. In addition, the application of JO-robot in polymerase chain reaction (PCR) has been explored by virtue of its versatile droplet manipulation capabilities. The extraction and purification of nucleic acids are realized. And the extracted nucleic acid templates are successfully amplified and analyzed. This multi-functional integrated and cross-scale droplet manipulation strategy gives the JO-robot great potential in microassays, medical diagnosis, fine chemical engineering, and microfluidic fields.

## Results

### Versatile droplet manipulation based on JO-robot

The schematic of JO-robot is shown in Fig. 1a. The top side of JO-robot is superhydrophobic with low droplet adhesion while the bottom side is hydrophobic with high adhesion. JO-robot tumbles directionally under the drive of the magnetic field and it can wrap the droplet with the assistance of capillary force and magnetic field. Two shallow creases are fabricated on the top surface of JO-robot for controllable folding. When the JO-robot comes into contact with the droplet, the droplet tends to reduce the liquid-air interface, thus lowering the surface energy. The capillary force generated in this process deforms the structure, thus increasing the elastic potential energy of the JO-robot, causing the robot to fold along the creases[25]. The magnetically controlled motion of the JO-robot allows both creases to contact the droplet and thus be folded. Compared with previous works that mainly focused on the capillary origami of nonresponsive substrates, such as molybdenum disulfide, polydimethylsiloxane, etc.[25,26], here, the magnetic substrate with the rational design of oriented magnetic particles chains and Janus wetting characteristic gives the origami structure diverse functionalities. After wrapping the droplet, the JO-robot can carry the droplet to tumble on the superhydrophobic substrate for directional transportation (Fig. 1a, b, Supplementary Movie 1). Compared with the droplet self-propelling induced by Laplace pressure difference[27], the droplet transport speed of JO-robot can reach as high as $68.47 \pm 2.1$ mm/s (6.5 μL droplet, Supplementary Fig. 1). Besides water droplet, liquids with high surface tension, such as liquid metal[28], can also be transported by JO-robot (Supplementary Fig. 2). By virtue of good magnetic response characteristic and untethered motion performance, the JO-robot further enables three-dimensional (3D) water droplet transportation. It can carry water droplet on vertical wall for anti-gravity climbing and can even transport droplet on inverted surface (Supplementary Fig. 3, Supplementary Movie 2). In addition to droplet transport, the JO-robot can also realize daughter droplet dispensing and release, droplet stirring and photothermal stirring, and droplet splitting (Supplementary Movie 3). Under the action of magnetic field and capillary force, the JO-robot is able to dispense the daughter droplet from the large droplet (Fig. 1c) and then release the wrapped droplet on-demand (Fig. 1d). It can also be used as a magnetic microstirrer to achieve micro-droplet stirring (Fig. 1e). In addition, it can be remotely heated by near-infrared laser (NIR laser), allowing simultaneous heating and stirring similar to the commercial magnetic mixer (Fig. 1f). The JO-robot also enables droplet splitting without any assistance. Unlike daughter droplet dispensing, splitting is the division of a large droplet into two similar smaller droplets (Supplementary Fig. 4).

### Characterization of the JO-robot

The JO-robot is constructed using polydimethylsiloxane (PDMS) doped with carbonyl iron particles, and it is fabricated by femtosecond laser writing and modification. With the advantage of efficient mask-less processing, the cutting of JO-robot contour, the creation of creases and the surface modification of the top side can be achieved in one step by the femtosecond laser processing, which is difficult to achieve by other magnetic microrobots fabrication methods[29–32] (Supplementary Fig. 5). The rectangular sheet-like soft structure has the advantages of simple design and various motion modes, and is widely used in the field of microrobots[33,34]. Here, the JO-robot is designed as a rectangle with two creases to achieve good magnetic response motion and multi-mode droplet manipulation (the size is 2 mm × 7 mm, the same hereinafter unless otherwise specified, detailed size is shown in Supplementary Fig. 6). During the curing process, the carbonyl iron particles doped in PDMS are arranged in chains to enhance the

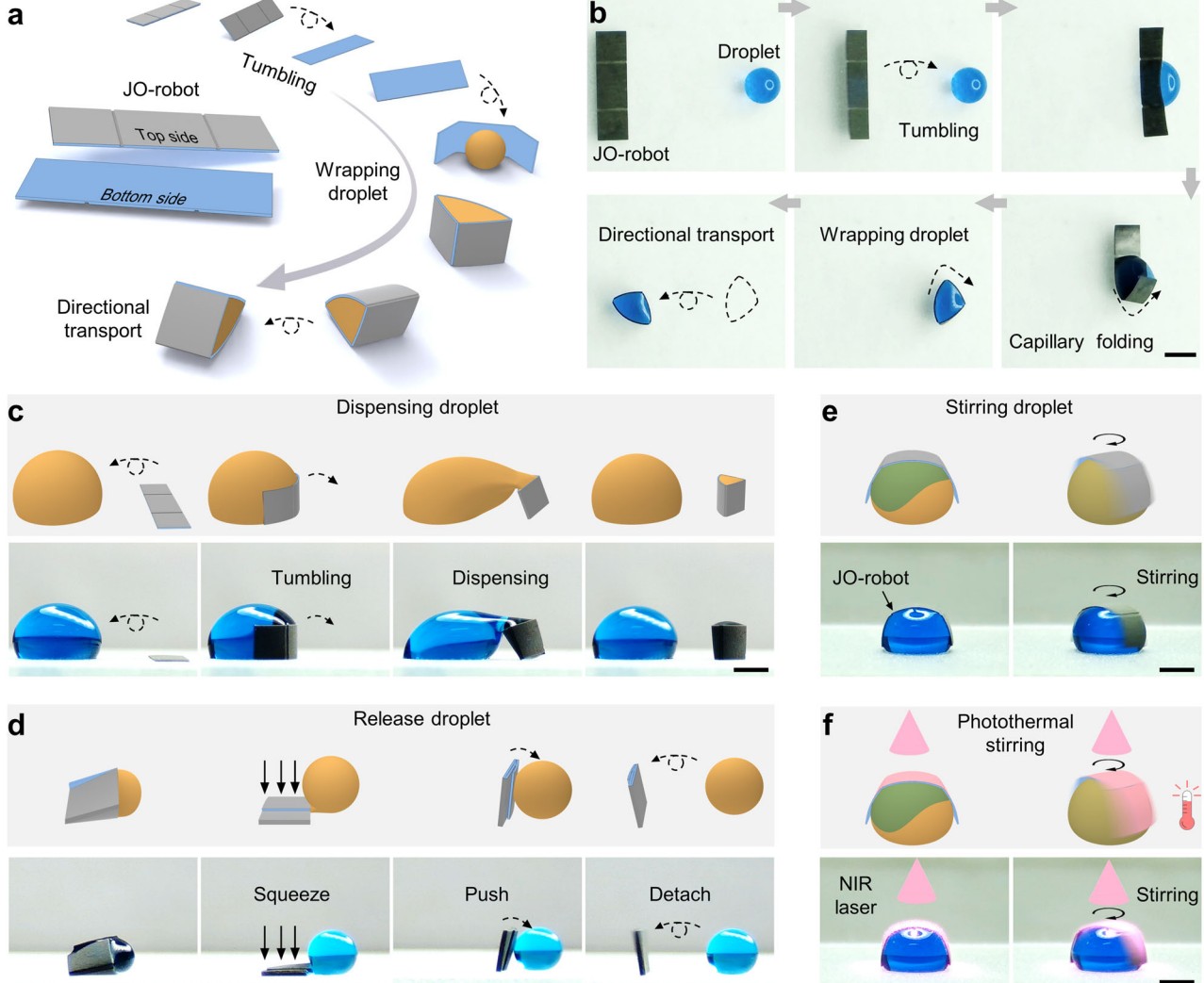

**Fig. 1 | Scheme illustration and demonstration of the JO-robot. a** Schematic of JO-robot. The top surface is superhydrophobic with low droplet adhesion and the bottom surface is hydrophobic with high droplet adhesion. With the help of capillary force, the JO-robot can wrap water droplet and transport droplet directionally by magnetically controlled tumbling. **b** The corresponding images of directional tumbling, wrapping droplet and directional transport of droplet by JO-robot. **c–f** Versatile droplet manipulation based on JO-robot. **c** The JO-robot enables daughter droplet dispensing under magnetic field and capillary force. **d** The wrapped droplet can be released by the JO-robot by squeezing and pushing. **e** The JO-robot can stir droplet under the action of the rotating magnetic field and **f** it can also realize photothermal stirring by NIR laser irradiation. All scale bars are 2 mm.

magnetic response. By controlling the laser cutting direction, the carbonyl iron particles chains are perpendicular to the long side of the JO-robot (Fig. 2a). The chains can be affected by the magnetic force and torque in the magnetic field, which drives the JO-robot to align with the direction of the magnetic field[35–37]. The magnetic force (**F**) and magnetic torque (**T**) excited by the periodic magnetic field can be given by (Supplementary Discussion 1):

$$\mathbf{F} = V_m(\mathbf{M} \cdot \nabla)\mathbf{B}(x, y, z) \qquad (1)$$

$$\mathbf{T} = V_m\mathbf{M} \times \mathbf{B}(x, y, z) \qquad (2)$$

Where $V_m$ is the volume of the JO-robot, **M** is the magnetization of JO-robot, **B**(x, y, z) is the magnetic flux intensity. When the chains are distributed perpendicularly to the long side of the robot, the tumbling axis overlaps with the long side. Therefore, the JO-robot can stably tumble around the long side to approach and manipulate the droplet under the action of the moving periodic magnetic field generated by magnets array (Supplementary Figs. 7, 8). In contrast, when the

carbonyl iron particles are uniformly distributed, the robot cannot be driven by the magnetic field (Supplementary Fig. 9). To facilitate folding and unfolding, two shallow creases with width of 138.1 µm ± 7.5 µm and optimized depth of 56.2 µm ± 3.9 µm are fabricated on the top surface (Fig. 2a). When the crease is too shallow, JO-robot is difficult to fold and can produce more residue when releasing the daughter droplet, while when the crease is too deep, it is difficult for JO-robot to unfold after releasing the droplet. The JO-robot is processed into a Janus structure by selective laser modification. The modified side is superhydrophobic (155.8° ± 1.4°) with low-adhesion while the unmodified side is hydrophobic (105.9° ± 2.1°) with high-adhesion (Supplementary Fig. 10). The scanning electron microscopy (SEM) images and 3D surface topography images clearly show the different topography of the two surfaces of the JO-robot (Fig. 2b). The modified surface is distributed with periodic bumped structures and covered with irregular micro-nanoparticles, while the unmodified surface is relatively flat. The lateral adhesion forces of the two surfaces are tested. The adhesion forces of superhydrophobic and hydrophobic surfaces are ~5.78 µN and ~67.54 µN, respectively (Fig. 2c). The JO-robot can tumble independently and can also carry water droplet for directional

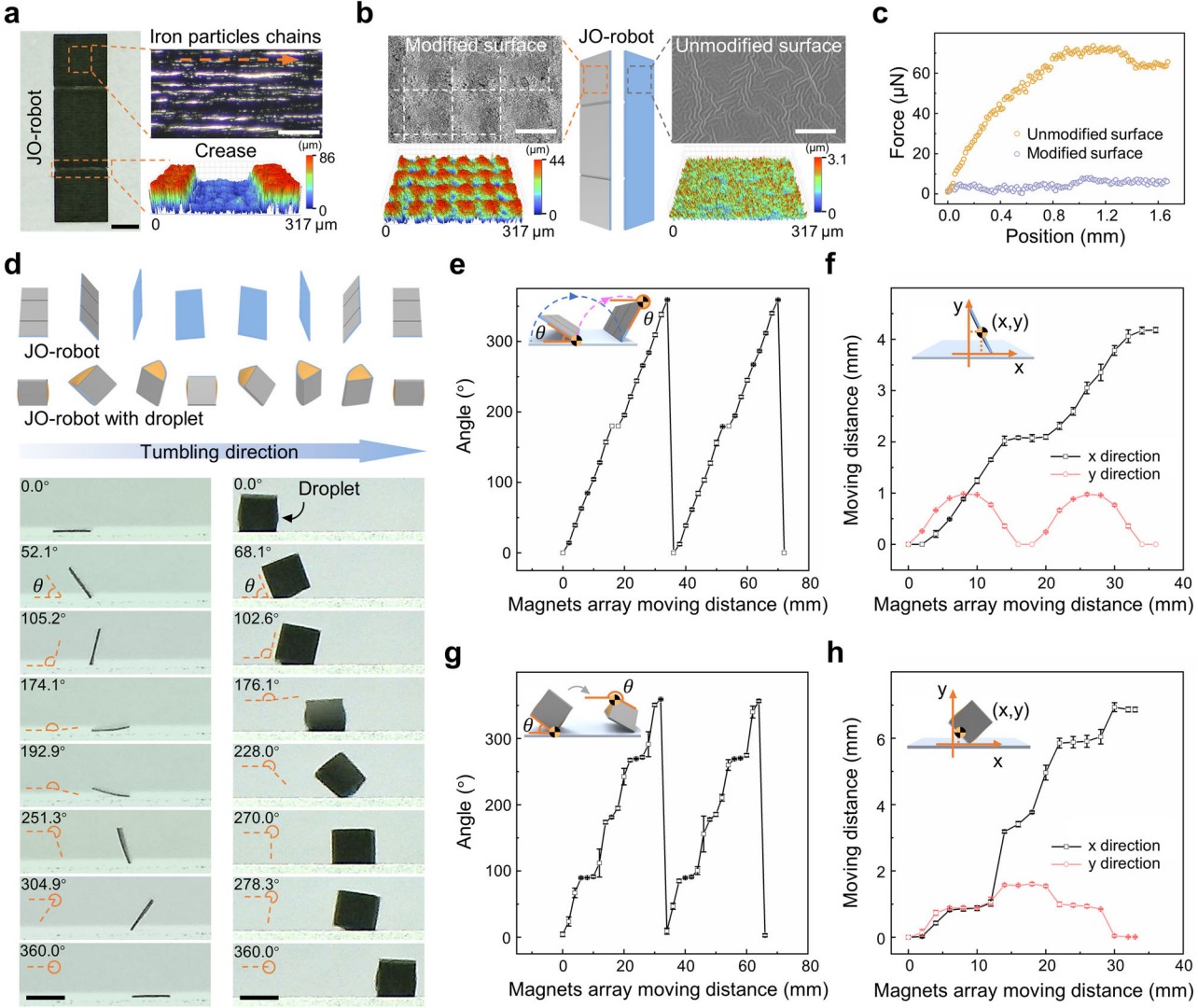

**Fig. 2 | Surface characteristics and magnetic response performance of the JO-robot. a** Images of the JO-robot. The carbonyl iron particles inside the JO-robot are arranged in chains. The chains are perpendicular to the long side of JO-robot, which makes the JO-robot to tumble around the long side. The 3D surface topography image shows the shallow creases on the JO-robot, which can guide the JO-robot to fold when wrapping the droplet. Scale bars are 1 mm (left) and 100 μm (right). **b** The SEM images and 3D surface topography images of the two surfaces of the JO-robot. The modified surface is rough and covered with microbumps while the unmodified surface is relatively flat. Scale bars are 50 μm. **c** The lateral adhesion force of the two surfaces of JO-robot (by sliding the droplet horizontally on the surface) **d** The schematic diagram and front view images of the movement of JO-robot and JO-robot with droplet in a tumbling cycle. Scale bars are 2 mm. **e-h** The tumbling angle and moving distance of (**e**), (**f**) JO-robot and (**g**), (**h**) JO-robot with droplet versus magnets array moving distance. The inset schematics show the definition of tumbling angle and moving distance. The error bars in (**e**), (**g**) represent the standard deviation of five independent measurements and the error bars in (**f**), (**h**) represent the standard deviation of three independent measurements. Source data are provided as a Source Data file.

transportation. The complete tumbling cycles are shown in Fig. 2d (Supplementary Movie 4). The tumbling angle and moving distance of JO-robot with and without droplet versus magnets array moving distance are shown in Fig. 2e–h. It shows that with the movement of the magnets array, the JO-robot can tumble from 0° to 360°.

**Daughter droplets dispensing and release**
Droplet dispensing and microdroplets array generation are key steps in microfluidic applications such as serial dilution and sample aliquoting[38,39]. The combination of origami deformation ability and the Janus wetting characteristic endows JO-robot with controllable daughter droplet dispensing and release abilities. As shown in Fig. 3a, the hydrophobic side of the JO-robot first contacts with the droplet. Under the action of the periodic magnetic field, the JO-robot starts to tumble around the side in contact with the substrate (edge I). As the JO-robot tumbles, it folds along the creases by the surface tension of the

liquid[25,40,41] and wraps a part of the liquid. With further tumbling of the JO-robot, the tumbling axis converts to the edge II. Both two sides of the JO-robot fold under the surface tension until the cross-section becomes a closed triangle. The folding time is managed by the moving speed of the magnets array and is not affected by the liquid surface tension, it can be controlled between ~68.75 ms and ~14.48 s (Supplementary Figs. 11, 12). As the JO-robot continues to tumble around the edge II, a portion of the wrapped liquid is pulled away from the large droplet with the formation of the necking point at the joint[11,42]. The formation and breakage of the necking point is the key to the precise dispensing of daughter droplets. With the tumbling of the JO-robot, the necking point becomes thinner and the radius of curvature of the necking point also decreases further. Finally, the necking point breaks under Laplace pressure, resulting in the dispensing of daughter droplet (Supplementary Movie 5)[11,42,43]. In addition to droplet dispensing, how to release the daughter droplets after dispensing remains a

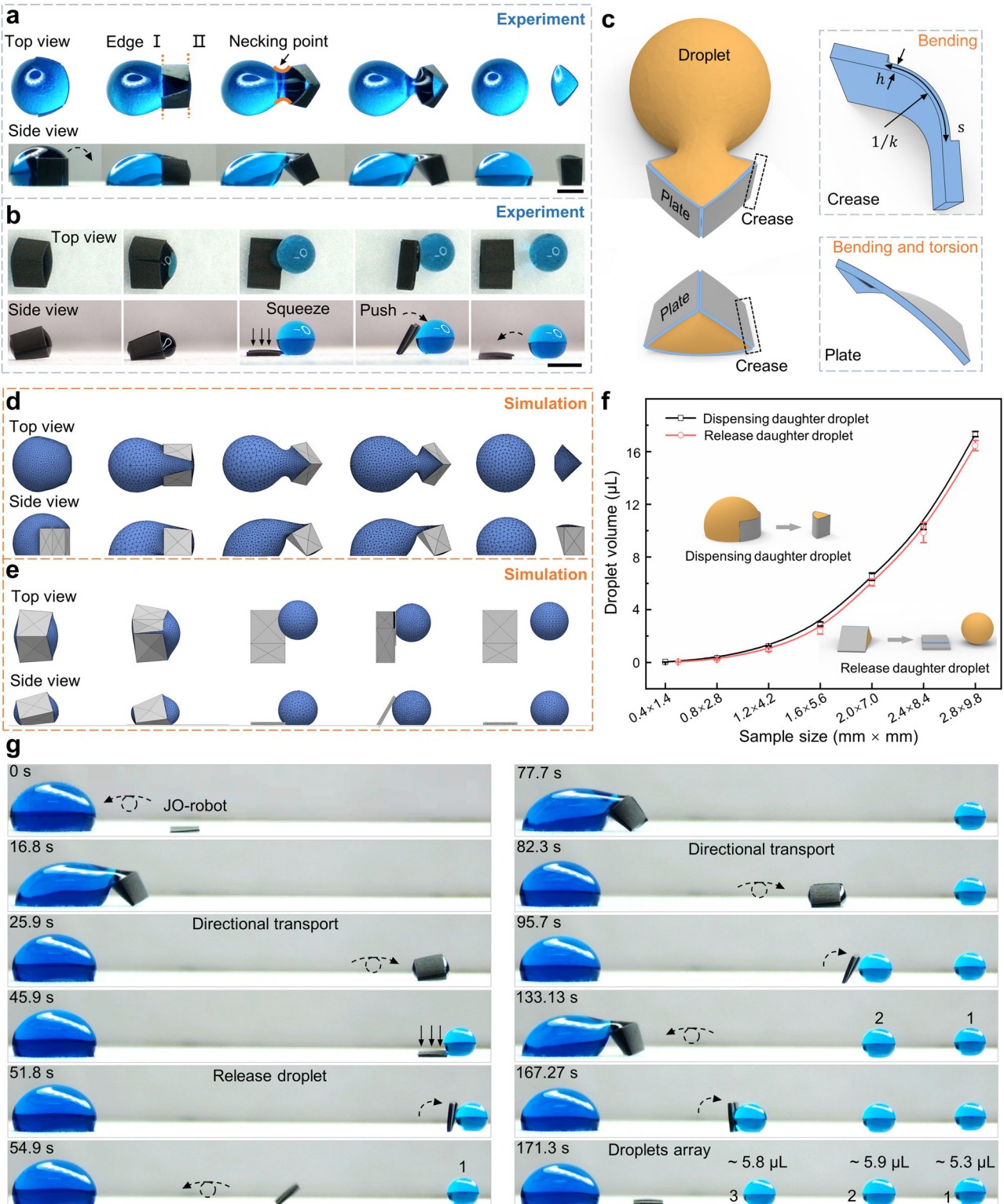

**Fig. 3 | JO-robot for daughter droplets dispensing and release. a** Controllable daughter droplet dispensing. **b** On-demand daughter droplet release. **c** Schematic diagram of JO-robot interaction with droplet during daughter droplet dispensing and release. The JO-robot is regarded as the three plates connected by creases. When interaction with droplet, the crease is bent (with the thickness of $h$, the arc length of $s$ and the curvature of $k$) and the plate is bent and twisted. **d**, **e** The simulations of the droplet morphology during dispensing and release, respectively. **f** The ability of JO-robots with different sizes to dispense and release daughter droplets. **g** On-demand generation of microdroplets array. Microdroplets array generation is achieved through continuous daughter droplets dispensing and releasing. All scale bars are 2 mm. The error bars represent the standard deviation of five independent measurements. Source data are provided as a Source Data file.

challenge for existing magnetic actuation droplet manipulation methods[12,15]. With the origami deformation ability and the Janus wetting characteristic, JO-robot can realize on-demand daughter droplet release. As shown in Fig. 3b, the JO-robot is first controlled to remain a slightly tilt to determine the direction of droplet extrusion. By increasing the magnetic field strength, the JO-robot can produce greater folding deformation, causing the droplet to be gradually squeezed out and eventually formed a spherical shape. It can autonomously release the daughter droplet by gently pushing the droplet through tumbling. In brief, on-demand daughter droplet release is achieved by deliberate squeezing and push-off motions (with the magnetic flux density larger than $265.0 \pm 24.7$ mT, $0.5 \pm 0.2$ mT and $0.6 \pm 0.2$ mT in the x-, y-, and z-directions, respectively. Supplementary Movie 6). During the dispensing and release processes, the Janus wetting characteristic of JO-robot plays the key role. The superhydrophobic outer surface is not wetted by droplet, which promotes the formation of the necking point, facilitates the dispensing of daughter droplet and avoids unnecessary liquid residue. It also ensures that the droplet does not stick to the JO-robot during release. Although other preparation methods such as 3D printing and ultraviolet lithography can easily prepare magnetic robots, they need additional surface modification to obtain Janus wetting characteristics[29,30]. The comparison tests of the droplet dispensing and release of magnetic robot without the Janus characteristic are shown in Supplementary Figs. 13, 14.

The equilibrium shape of the droplet during dispensing and release can be obtained from energy principles[43,44]. As shown in Fig. 3c, the JO-robot can be regarded as the three plates connected by creases. During the interaction with droplet, the crease is bent (with the thickness of $h$, the arc length of $s$ and the curvature of $k$) and the plate is bent and twisted. The total system energy can be given by (Supplementary Discussion 2[44,45]):

$$E_t = E_k + E_\gamma + E_g + E_m \qquad (3)$$

where $E_t$ is the total energy of the system, $E_k$ are the strain energies of the creases bending and the plates bending and torsion, $E_\gamma$ are the interface energies, $E_g$ are the gravitational energies of JO-robot and water droplet, and $E_m$ is the magnetic energy. The total energy is minimum when the system is in equilibrium. In order to study the equilibrium shapes of the droplet, we use the Surface Evolver finite element program to perform simulations, which is realized by minimizing the total system energy[46]. During the dispensing and release process, the capillary number $Ca = \mu U/\gamma \sim 10^{-5} \ll 1$ and the Weber number $We = \rho U^2 L/\gamma \sim 10^{-6} \ll 1$ (where the $\mu$, $U$, $\gamma$, $\rho$, $L$ are the viscosity, velocity, surface tension, density and characteristic length of the droplet, respectively), so the system satisfies the quasi-static assumption[43]. In the simulation, the droplet is first divided into small triangular meshes. The equilibrium shape of the droplet is obtained by refining the triangular mesh and moving vertices for several iterations to minimize the total potential energy of the system under the constraints of volume and contact angle. The simulated results are shown in Fig. 3d, e.

To investigate the scalability of this droplet manipulation strategy, JO-robot of different sizes are prepared for daughter droplet dispensing and release (Fig. 3f). The relationship between the designed dimensional parameters and the volume of the dispensed and released droplets is shown in Supplementary Discussion 3. Daughter droplets from 28 nL $\pm$ 4.4 nL to 17.34 μL $\pm$ 0.2 μL can be dispensed. The minimal daughter droplet that can be successfully dispensed by JO-robot (0.23 mm $\times$ 0.94 mm) can reach ~3.2 nL (Supplementary Fig. 15, Supplementary Movie 7). And the maximum dispensable daughter droplet is ~51.14 μL (the size of JO-robot is 4.03 mm $\times$ 14.44 mm). The daughter droplets from 42.21 nL $\pm$ 9.5 nL to 16.46 μL $\pm$ 0.39 μL can be released by the JO-robot with different sizes. The minimum volume of the releasable droplet is ~30.2 nL (the size of JO-robot is 0.5 mm $\times$ 1.75 mm). In addition, we further discuss the effects of different design parameters on droplet dispensing and release, such as the orientation of carbonyl iron particles chains, the number and distribution of creases, and the width and length of the JO-robot (Supplementary Figs.16–23). With these daughter droplet dispensing and release capabilities, the JO-robot can realize on-demand generation of microdroplets array, which is incapable with other magnetic actuation droplet manipulation strategies (Fig. 3g, Supplementary Movie 8). First, the JO-robot dispenses daughter droplet 1 from the large droplet and transports it directionally to the target location for release. Then the JO-robot returns to the large droplet to dispense and release the daughter droplet 2 in the same manner. The on-demand generation of microdroplets array can be achieved by repeating the above steps. Here, three daughter droplets (droplets 1–3) are demonstrated as an example. The volume of generated daughter droplets 1–3 are ~5.3, ~5.9 and ~5.8 μL, respectively. It indicates the uniformity and reliability of the JO-robot for droplet dispensing. The magnetic flux density required for droplet manipulation including the tumbling of JO-robot, the dispensing and transportation of droplet is between 81.5 $\pm$ 5.7 mT and 155.4 $\pm$ 10.4 mT (the magnetic field information is shown in Supplementary Fig. 24 and Supplementary Table 3). In addition to squeezing out droplet, the JO-robot can also realize long-distance (~33.04 mm) and high-speed (~75.09 mm/s) droplet ejection through simple surface modification (Supplementary Fig. 25, Supplementary Movie 9).

## Droplet stirring, photothermal stirring and versatile droplet manipulation integration

Stirring is one of the active ways to achieve rapid mixing of liquids for effective biological and chemical reactions[47,48]. JO-robot can rotate on the droplet, allowing rapid stirring of droplet for efficient mixing (Fig. 4a). It is driven by the magnetic torque ($\mathbf{T}_m$) generated by the rotating magnetic field, which can be given by (Supplementary Discussion 1):

$$\mathbf{T}_m = V_m \mathbf{M} \times \mathbf{B}_c(x, y, z) \qquad (4)$$

where $\mathbf{B}_c(x, y, z)$ is the magnetic flux intensity generated by the cube magnet. The velocity and flow behavior of the liquid during stirring are studied by the droplet internal flow field simulation (Fig. 4b). It shows that the liquid at the edge away from the rotation axis has the fastest movement speed (~52.86 mm/s, the angular speed of JO-robot is $8\pi$ rad/s). In practical tests, the stirring process in the droplet can be monitored by the fluorescein (Fig. 4c, Supplementary Movie 10). The 6.5 μL droplet containing 0.01 M fluorescein sodium salt is added to the sessile droplet (30 μL) from the top. As can be seen from the fluorescence images, the fluorescence spreads to the whole droplet within one second with the rotation of the JO-robot (magnetic flux density is 25.85 $\pm$ 0.7 mT). A more homogeneous mixing is achieved around the fifth second. Normalized two-dimensional fluorescence distribution maps are used to show the mixing more intuitively. It shows that the fluorescence intensity inhomogeneity in the droplet decreases gradually within five seconds. In contrast, the homogeneous mixing by passive diffusion takes about ~100 s (Supplementary Fig. 26, Supplementary Movie 10). The mixing efficiency is increased by about 20 times through the rotation of the JO-robot, and it can further be improved by enhancing the rotation speed of JO-robot. In addition to the conventional droplet manipulation functions, photothermal property of the JO-robot is also explored to achieve remote heating of droplet while stirring. Carbonyl iron particles in JO-robot can efficiently absorb and convert near-infrared (NIR) laser into heat[37]. Based on this photothermal property, the JO-robot can simultaneously heat and stir droplet (Fig. 4d). In viscous liquids, such as glycerol, diffusive (passive) mixing of solutes is difficult and inefficient mainly

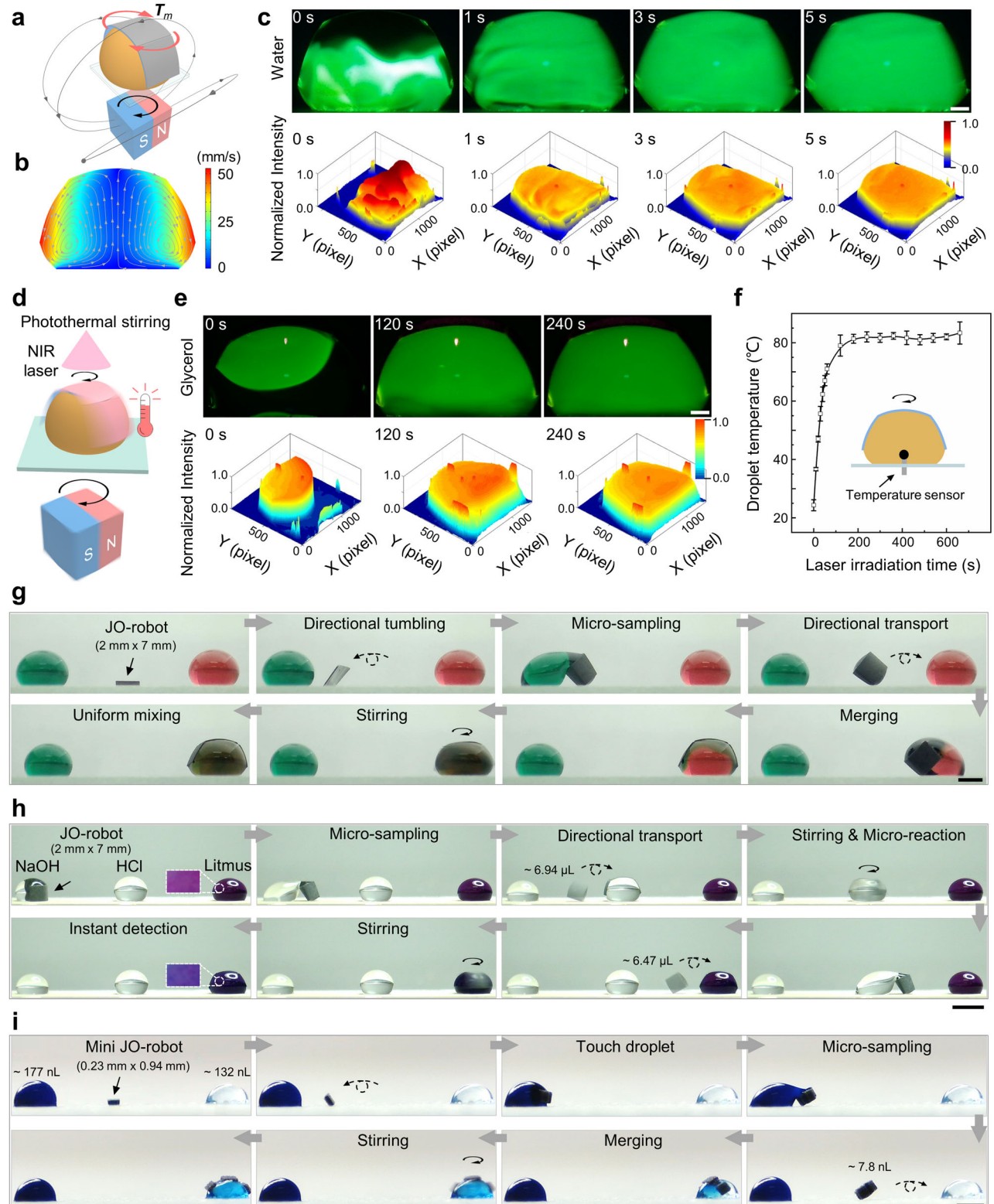

because of the laminar nature of flow at low Reynolds numbers[49]. By integrating the photothermal function, the JO-robot can be used to heat glycerol during mixing to reduce the viscosity of glycerol and thus improve mixing efficiency. The 6.5 μL glycerol droplet containing 0.01 M fluorescein sodium salt is added to the 30 μL sessile glycerol droplet from the top. For better demonstration of the photothermal effect, JO-robot rotates with a low rotation speed of 1.14π rad/s. As shown in Fig. 4e, with the remote heating of the NIR laser, the glycerol

can be gradually mixed by the JO-robot in 240 s (Supplementary Movie 11). Normalized two-dimensional fluorescence distribution maps show clearly that the fluorescence intensity inhomogeneity gradually decreases within 240 s with the hearting and stirring. The temperature of the glycerol increased rapidly from 24.3 to 81.3 °C in 180 s during the mixing process and finally stabilized at about 82 °C (Fig. 4f). However, in the absence of NIR light heating, the mixing of glycerol under the same condition takes about 30 min (Supplementary

**Fig. 4 | Droplet stirring, photothermal stirring, and versatile droplet manipulation integration based on JO-robot. a** Schematic of the JO-robot stirring water droplet under the rotating magnetic field. The rotating magnetic field is generated by rotating the cube permanent magnet (25 mm × 25 mm × 25 mm) by the stepping motor. **b** The simulation of droplet internal flow field. **c** Time lapsed fluorescence images and normalized two-dimensional fluorescence distribution maps during droplet stirring. Scale bar is 500 μm. **d** Schematic diagram of the photothermal stirring. **e** Time lapsed fluorescence images and normalized two-dimensional fluorescence distribution maps during photothermal stirring of glycerol droplet. Scale bar is 500 μm. **f** The temperature curve of glycerol droplet during photothermal stirring. The inset shows the position of the temperature sensor during temperature detection. **g** The versatile droplet manipulation integration of JO-robot. Two dyed aqueous glycerol solutions (50% v/v) are used as demonstration. Scale bar is 2 mm. **h** The versatile droplet manipulation integration of JO-robot for microchemical reactions including micro-sampling, sample transportation, sample addition, rapid reaction and instant detection. Scale bar is 4 mm. **i** Versatile manipulation of nL-scale droplets based on the mini JO-robot, including droplet dispensing, transport, merging and mixing. Scale bar is 500 μm. The error bars represent the standard deviation of five independent measurements. Source data are provided as a Source Data file.

Fig. 27, Supplementary Movie 11). In addition to promoting viscous fluid mixing, this photothermal mixing capability of the JO-robot has the potential to be used for microchemical reactions and biological applications[50,51].

By integrating the stirring function with the droplet dispensing and transport functions, the JO-robot can effectively realize the continuous droplet manipulation objectives. As shown in Fig. 4g, JO-robot first contacts the green droplet by tumbling, and then samples a portion of the liquid. After that, JO-robot directionally transports the sampled green daughter droplet through tumbling motion and merges it with the red droplet. Finally, JO-robot uniformly mixes the combined droplet by rotation (Supplementary Movie 12). Micro-sampling, sample transportation and addition, rapid reaction and products instant detection are promising in microchemical reactions and biological applications. A simple chemical reaction is used as an illustration (Fig. 4h, Supplementary Movie 13). First, the JO-robot dispenses a ~6.94 μL sodium hydroxide (NaOH) droplet from the large droplet (~30 μL), and then transports and merges it with a ~30 μL hydrochloric acid (HCl) droplet. The merged droplet is fully stirred by the JO-robot to ensure the complete reaction. The reacted droplet remains transparent without any precipitation or gas production, making it difficult to directly judge the character of the reacted solution. Therefore, it is necessary to use the JO-robot to further dispense from the reacted droplet and mix the sample (~6.47 μL) with the detection reagent Litmus (~30 μL) for instant detection. As shown in Fig. 4h, the Litmus reagent changes from purple to blue after sufficient stirring, indicating that the reacted liquid is alkaline. This proves that the concentration of NaOH solution is higher than that of HCl solution. In addition, the JO-robot can be detached from the droplet in two ways after completing the manipulation targets (with and without folding, Supplementary Figs. 28, 29, Supplementary Movie 14). Reducing the droplet size to the nL-scale causes manipulation difficulties. Although already applied to μL-scale droplets, the versatile manipulation of nL-scale droplets is still challenging for existing magnetic actuation droplet manipulation strategies[9,10,23,24]. Here, the scalable JO-robot-based droplet manipulation strategy enables versatile manipulation integration of nL-scale droplet, including daughter droplet dispensing, transport, merging and mixing. A mini JO-robot with the size of 0.23 mm × 0.94 mm is used to dispense a nL-scale daughter droplet (~7.8 nL) from the dyed droplet (~177 nL) and directionally transport the daughter droplet to another droplet (~132 nL). After merging, the rapid stirring of the combined droplet can also be achieved by the mini JO-robot for efficient homogeneous mixing (Fig. 4i, Supplementary Movie 15).

## Nucleic acids extraction and purification

The versatile droplet manipulation capabilities of the JO-robot are further explored for biochemical applications. Since the outbreak of the epidemic coronavirus disease (COVID-19), rapid and high sensitivity laboratory diagnosis such as the viral nucleic acid detection based on polymerase chain reaction (PCR) plays an important role in disease prevention and control[52,53]. To execute the nucleic acid detection, nucleic acids should first be extracted from the lysed biological samples (blood, saliva, semen, etc.). By coating the pH-responsive material on the silica-based sorbents, the nucleic acids can be captured in the moderately low pH environment and released in the moderately high pH environment, so as to realize the extraction and purification of nucleic acids[54,55]. Chitosan is a particularly useful pH-responsive material for PCR application[54,56-59], which is a naturally occurring cationic polysaccharide with abundance amine groups. The amine groups can be charge-regulated by pH with a pKa of about 6.3. The protonation of the amine group in acidic solution (below 6.3) makes the chitosan cationic which can bind the anionic nucleic acid by electrostatic adsorption from the biological samples. At a pH value around 8.5, the chitosan becomes nearly neutral, leading to the effective elution of nucleic acids[54,56-59].

Here, the chitosan-modified JO-robot is used for on-chip nucleic acid extraction and purification. To enable effective adsorption and elution of nucleic acids, chitosan is used to modify the JO-robot surface. First, the laser modified surface of the JO-robot is activated by the oxygen plasma for 1 min[56]. After that, the JO-robot is soaked in chitosan solution for 18 h. During the chitosan modification process, the plasma-treated side is downward to contact with the solution and the other side is upward to maintain contact with air, so that only one side can be modified by chitosan. Then, the JO-robot is washed three times with deionized water (Fig. 5a). As shown in Fig. 5b, the chitosan-coated JO-robot can be used to bind and release nucleic acid. The chitosan is cationic at pH = 5 and can effectively bind the anionic nucleic acid from the complex biological samples by high electrostatic adsorption. With the increase of pH value, the chitosan becomes close to neutral at pH = 9, causing the release of nucleic acids. The extraction and purification of nucleic acids using JO-robot are performed in a mineral oil environment to prevent evaporation of droplets[60] and they can be achieved through several steps (Fig. 5c). (i) The chitosan-modified side on JO-robot is first pre-wetted with lysis buffer (pH = 5). And then the JO-robot is placed on the surface of the lysed sample (pH = 5). The pre-wetted side is underoil superhydrophilic and the other side is superhydrophobic[61] (Supplementary Fig. 30). So, the JO-robot can spontaneously float at the water-oil interface with the chitosan-modified side facing the reagent. A magnetic field is used to control the rotation of the JO-robot (120 rpm for 5 min) so that it can make full contact with nucleic acids (DNA) for adsorption. (ii) The nucleic acid-bound JO-robot is carefully slid out along the water-oil interface without folding. Unlike the tumbling and droplet dispensing processes of the JO-robot controlled by the magnets array, the sliding out process is controlled by a single magnet and it is similar with the sliding out process in air (Supplementary Fig. 29). First, the JO-robot that suspended at the top of the droplet moves to the side of the droplet as the magnet approaches vertically from below. As the magnet gets closer, the JO-robot is subjected to a greater magnetic action than capillary action, so that the plates on both sides of the robot can overcome capillary folding to unfold. As the magnet moves to the right, the JO-robot tilts to the left. The angle between the JO-robot and the substrate decreases. As the magnet continues to move right, the JO-robot is finally attracted by the strong magnetic field and slides out along the contour of the droplet without folding. Then it is controlled to tumble towards the next reagent (washing buffer 1). (iii) The nucleic

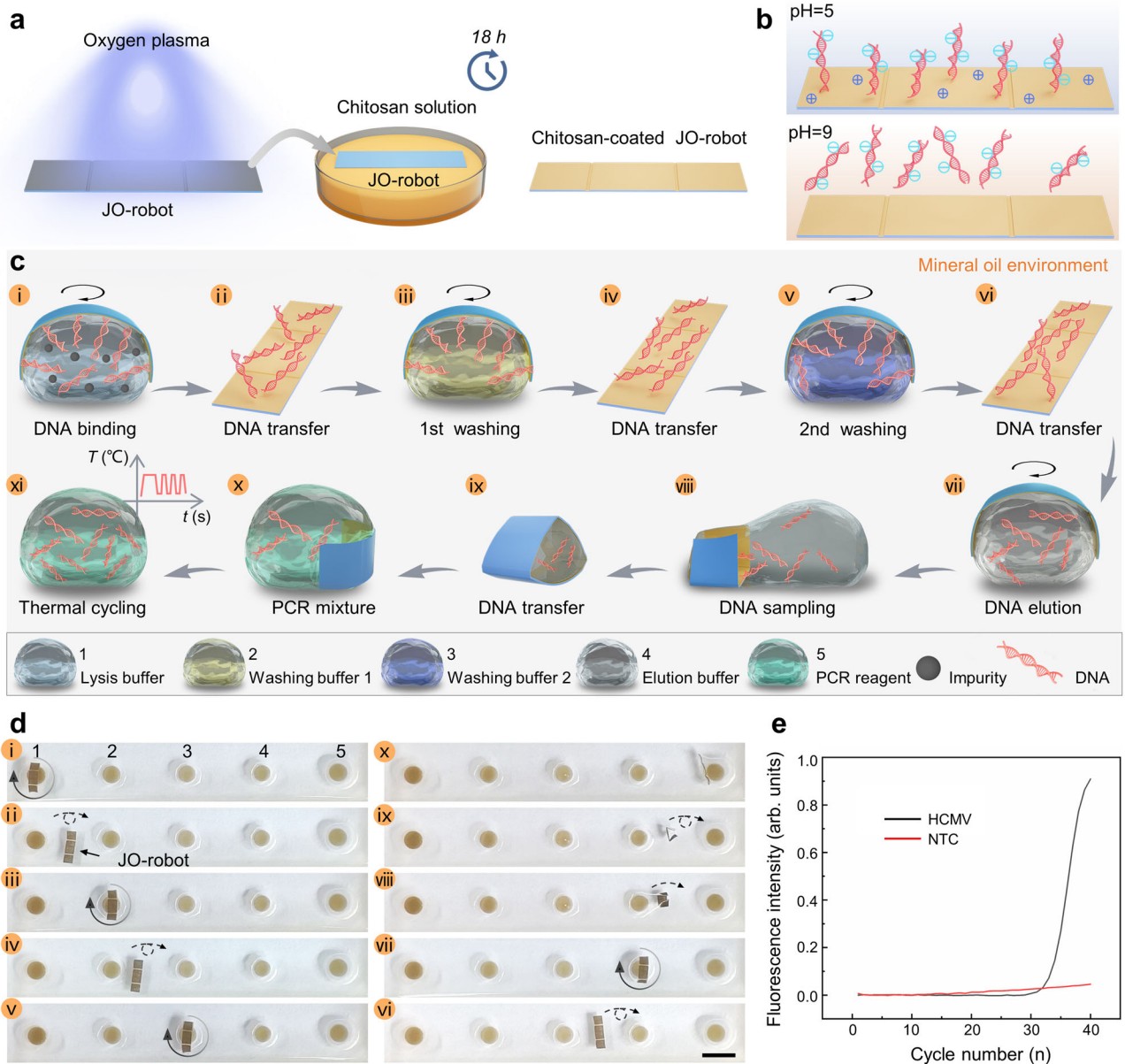

**Fig. 5 | Nucleic acids extraction and purification based on JO-robot. a** Schematic diagram of surface modification of JO-robot with chitosan. **b** Schematic of the nucleic acids binding and release mechanism of chitosan-coated JO-robot. Chitosan is cationic at pH = 5 and can bind the anionic nucleic acid by high electrostatic adsorption. Whereas chitosan becomes close to neutral at pH = 9, leading to the release of nucleic acid. **c**, **d** Schematic diagram and images of the on-chip nucleic acids extraction and purification using chitosan-coated JO-robot, respectively. Scale bar is 5 mm. **e** The amplification curve of HCMV. Source data are provided as a Source Data file.

acid-bound JO-robot moves to the washing buffer 1. It can still spontaneously float on the water-oil interface with the nucleic acid-bound side (the chitosan-modified side) facing the washing buffer 1. And then, the JO-robot rotates to wash and remove impurities (120 rpm for 30 s). (iv) JO-robot transfers nucleic acids to the next reagent. (v) The same as procedure (iii), the nucleic acid-bound JO-robot is washed again in washing buffer 2. The purpose of twice washes is to remove impurities such as lipids and proteins to retain pure nucleic acids. (vi) Nucleic acids are transferred again. (vii) JO-robot transports to elution buffer and rotates to fully elute nucleic acids (120 rpm for 5 min). (viii) Using the daughter droplet dispensing function of JO-robot to precisely dispense eluted nucleic acids (-2 μL). (ix) Transfer the sampled nucleic acids. (x) Mixing the nucleic acid templates with PCR reagents. (xi) A portion of the PCR mixture (10 μL) is collected for PCR amplification and analyzed on a quantitative PCR instrument.

The images of the nucleic acid extraction and purification are shown in Fig. 5d. To enable miniaturized nucleic acid extraction and purification, a chip composed of silicone frame and super-hydrophobic glass substrate with local hydrophilic points is first prepared (Supplementary Fig. 31). Droplets 1–5 are added to the chip (local hydrophilic points) in sequence, which are the lysed sample (50 μL), washing buffer 1 (20 μL), washing buffer 2 (20 μL), elution buffer (20 μL) and PCR reagent (43 μL), respectively. Finally, the mineral oil is added to the chip to completely cover the droplets. Here, the 45 μL reaction volume including 2 μL template and 43 μL PCR reagent is used. For precise dispensing of 2 μL template in the step viii, the size of JO-robot is carefully designed (1.52 mm × 5.24 mm). And the optimized JO-robot can realize precise dispensing of droplet with a volume of 2.03 μL ± 0.37 μL in the mineral oil environment. In addition, microcolumn arrays are prepared on the surface of the JO-robot to

increase the nucleic acids binding area (Supplementary Fig. 32). Nucleic acids extraction and purification using the JO-robot can reduce residuals from the previous reagent and facilitate accurate dispensing of eluted nucleic acid sample with its droplet dispensing capabilities. The nucleic acids extraction performance of JO-robot is verified by detections of simulated samples of human cytomegalovirus (HCMV)[62,63]. And the nucleic acids extraction and purification steps are consistent with steps i to xi as mentioned above (Supplementary Movie 16). The amplification curve of HCMV is shown in Fig. 5e. It can be seen that the nucleic acids obtained using the proposed scheme are successfully detected. The corresponding cycle threshold (Ct) value is 32, and no obvious signal is identified in NTC (no-template control), indicating that the chitosan-modified JO-robot is capable of extracting and purifying the nucleic acids.

## Discussion

In conclusion, a full landscape of droplet manipulation strategy based on Janus origami robot is showcased. The JO-robot wraps and releases droplets with capillary forces that interact with droplets. It can be controlled by magnetic fields to tumble, fold and rotate for versatile droplet manipulation behaviors including 3D transportation, merging, splitting, on-demand daughter droplet dispensing and release, stirring and remote heating. The high stability of this strategy confers the ability to manipulate cross-scale droplets. Versatile droplet manipulation can be performed for droplets ranging from ~3.2 nL to ~51.14 μL. Compared with other magnetic actuation droplet manipulation strategies, the combination of origami deformation ability and Janus wetting characteristic endows JO-robot with higher flexibility and wider droplet volume applicability, while avoiding the mixing of droplets with magnetic additives, simplifying the droplet manipulation steps and avoiding the risk of droplet contamination. With its versatility and cross-scale applicability, the JO-robot can work as a precision manipulator to maneuver droplets on demand, thus being of considerable significance for fine chemistry, medical diagnostics and microfluidics where precision acquisition and addition of reagents, microdroplet patterning and rapid microdroplet reactions are widely required.

## Methods

### Materials

Droplet manipulation is performed on the superhydrophobic substrate, which is prepared by spraying the frosted glass with commercial superhydrophobic spray (Glaco Mirror Coat Zero, Soft 99 Co.). The dyed droplets are prepared by adding methylene blue ($C_{16}H_{18}ClN_3S\cdot3H_2O$) to deionized water for droplet manipulation. Fluorescein sodium salt ($C_{20}H_{10}Na_2O_5$) is added to deionized water and glycerol ($C_3H_8O_3$) as the fluorescent indicator for liquid mixing. And fluorescence is excited by using a light emitting diode (LED) lighting system with wavelengths of 450–460 nm. The light emitted from the droplet is filtered using a green optical filter. The NIR light with wavelength of 808 nm and power of 430 mW (spot size of 7 mm × 5 mm and irradiation distance of 107 mm) is used to irradiate the JO-robot for photothermal stirring.

### Characterization

The surface morphology of JO-robot is characterized by the scanning electron microscope (ZEISS EVO18) and 3D optical profiler (ContourGT-K, Bruker Nano GmbH, Berlin, Germany). The optical images are taken by the charge-coupled device camera. The contact angle of JO-robot is measured by the contact angle meter (CA-100C, Shanghai innuo precision instruments Co., Ltd). The volume of the test droplets is 4 μL. The average contact angle values are obtained by taking at least five measurements at different positions of the JO-robot. The adhesion force is measured by a customized surface force meter. The magnetic field simulation and flow field simulation are performed by COMSOL

software. The droplet morphology simulation is obtained by using SURFACE EVOLVER software.

### Fabrication of magnetic film

The carbonyl iron powder with diameter of 3–5 μm (≥99.5% purity, Nangong Rui Teng Alloy Material Co., Ltd.) and polydimethylsiloxane (PDMS, Sylgard 184, Dow Corning) are used to prepare the magnetic film. First, the liquid PDMS prepolymer, carbonyl iron powder and cross-linker are thoroughly mixed in a mass ratio of 10:4:1 to prepare the iron powder-doped PDMS. The mixture is then placed in a vacuum chamber and fully degassed to remove air bubbles. After degassed, the carbonyl iron powder-doped PDMS is poured onto the glass slide for spin-coating. Before use, the glass slide is treated with a super-hydrophobic spray (Glaco Mirror Coat Zero, Soft 99 Co.) to facilitate the detachment of the prepared JO-robot. After spin-coated (1200 rpm, 15 s, SpinCoater, Opticoat MSA100, Mikasa Co. Ltd., Japan), the sample is placed in a directional magnetic field generated by two neodymium-iron-boron (NdFeB) permanent magnets (100 mm × 50 mm × 10 mm) to align the carbonyl iron particles in the liquid PDMS into chains along the direction of the magnetic field. Finally, the magnetic film (95.9 ± 3.5 μm) can be obtained by thermal curing (100 °C, 30 min).

### Femtosecond laser fabrication

The femtosecond laser (pulse width <100 fs, repetition rate of 1 kHz, central wavelength of 800 nm) is used to cut and modify the magnetic film to prepare the JO-robot. The laser beam generated by the Ti:sapphire femtosecond laser system (80L8TICEACE-100F-1K, Spectra-Physics Solstice Ace, USA) is guided into the scan system (s-9210d, Sunny Technology, China) and focused onto the magnetic film through the F-theta lens (focal length ~ 63 mm). The sequence and speed of the scanning path can be controlled by the scan system. The outer contour (300 mW, 10 mm/s, 14 circles) and creases (300 mW, 50 mm/s, 14 circles) of the JO-robot and the modification of the top surface (300 mW, 50 mm/s, 3 circles, grid scanning at 60 μm intervals) are processed in one step by laser scanning. After standing overnight, the top surface can have superhydrophobic and low adhesion properties. JO-robot can be obtained by carefully removing the processed sample from the slide.

### PCR application

The buffers for nucleic acids extraction and purification are obtained from commercially available kit (DP322, TIANamp Swab DNA Kit, TIANGEN BIOTECH Co., Ltd.). The pH value of the lysis buffer and elution buffer is adjusted with HCl and NaOH respectively. The HCMV-positive quality control (DA0121, Diagnostic kit for Quantification of Human Cytomegalovirus DNA (PCR-Fluorescence), HCMV AD-169 standard strain dilution, $1 \times 10^6$ copies/mL, 2.5 μL, DaAn Gene Co., Ltd.) is mixed with the oral secretions to simulate the real HCMV sample. And the PCR reagent is obtained from the Diagnostic kit for Quantification of Human Cytomegalovirus DNA (PCR-Fluorescence) (DA0121, DaAn Gene Co., Ltd.). The simulated HCMV sample is mixed with the lysis buffer (52.5 μL) for cell lysis. A portion of the lysed sample (50 μL) is used for nucleic acid extraction. To enable effective adsorption and release of nucleic acid, chitosan (Sigma Aldrich, St. Louis, MO) is used to modify the JO-robot surfaces. First, the low molecular weight chitosan (0.2 g) is dissolved in 1% acetic acid solution (10 ml) to prepare the chitosan solution. And then, the oxygen plasma-treated JO-robot (75 W and 60 Pa for 60 s, Mingheng PDC-MG, China) is placed on the surface of chitosan solution for modification (the treated side contacts the solution and the untreated side contacts the air). The nucleic acid extraction and purification are carried out in a sealed mineral oil (M5904, Sigma Aldrich, St. Louis, MO) environment. The PCR amplification and detection of obtained nucleic acid is performed on the Roche Light Cycler 96 system (Basel, Switzerland). The PCR

amplification procedures are set at 93 °C for 120 s, followed by 40 cycles of 93 °C for 5 s and 57 °C for 45 s, and lastly at 37 °C for 30 s. According to the kit instructions, the samples are considered positive when presenting a typical amplification curve with a Ct value <37.

## Data availability

All data needed to evaluate the conclusions in the paper are present in the manuscript and Supplementary Information. The data are also available upon request from the corresponding author. Source data are provided with this paper.

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

## Acknowledgements

This work was financially supported by the National Natural Science Foundation of China (Nos. 52122511, 61927814, 52075516, U20A20290), the National Key Research and Development Program of China (No. 2021YFF0502700), Major Scientific and Technological Projects in Anhui Province (202203a05020014), Youth Innovation Promotion Association CAS (Y2021118), and the Research Grants Council of Hong Kong (GRF 17205421, 17204420, 17210319). We acknowledge the Experimental Center of Engineering and Material Sciences at USTC for the fabrication and measuring of samples. This work was partly carried out at the USTC Center for Micro and Nanoscale Research and Fabrication. The authors acknowledge Associate Professor Yunlong Jiao from Hefei University of Technology for his help in the surface adhesion testing.

## Author contributions

S.J., Y.H., and D.W. conceived the idea and designed the project. S.J., B.L., H.Y., and B.D. carried out the mechanical analysis and the simulations. S.J., J.Z., and L.Z. designed and completed PCR application. Y.Z., C.Z., and R.L. performed the characterization. S.J., K.S., J.H., and Z.S. fabricated samples. S.J., Y.Z., Z.Z., and C.C. analyzed the data and prepared the figures. S.J., D.W., L.Z., J.L., L.W., J.C., and Y.H. wrote and revised the manuscript.

## Competing interests

The authors declare no competing interests.
