## [Peer Review File · Nature Communications]

REVIEWER COMMENTS

Reviewer #1 (Remarks to the Author):

The paper describes a multi-scale origami robot(exosuit) capable of cross-scale droplet manipulation with magnetic actuation. The results show that the origami robots perform various versatile droplet manipulation including three-dimensional transport, merging, splitting, releasing, stirring and remote heating of droplet of multiple size. The reviewer hopes the paper can be improved by addressing following comments.

- In title, reviewer recommends replacing the name of the device (Exosuit) with “robot” or another word that can represent the device properly. “Exosuit” may mislead readers. It’s just recommendation. I understand why the authors name the device as an exosuit, but exosuit is for wearing not manipulating.

- The authors show many functional capabilities of the proposed device. It would be a much more worthwhile paper, if the authors give a design principle for this droplet manipulation. For example, how to manage droplet volume in separation, relationship between carrying volume of droplet and design parameters of the device. If it is out of focus in this research, it would be helpful to provide discussion about main design parameters of the proposed device such as dimensional size, material treatment, shape of the device etc..

- Except for the functional performance of the device, could you summarize enabling technologies or key design factors for manipulating droplet with magnetic origami? It is helpful to understand why existing technology couldn’t realize these functions. Please compare with following paper.

Tianqi Xu et al. “Millimeter-scale flexible robots with programmable three-dimensional magnetization and motions”, Science robotics, 2019

Reviewer #2 (Remarks to the Author):

The manuscript by Jiang et al. reports on the three-dimensional manipulation of droplets utilizing millimeter scale origami-based robot assisted by conjunction of magnetic force and capillary force. Authors nicely demonstrated dispensing, releasing, stirring, and mixing droplets by designing robots with aligned magnetic particles and Janus wettability of superhydrophobic and hydrophobic properties. Especially, extraction and purification of nucleic acids is an outstanding demonstration by

untethered stirring and dispensing of the droplets. However, there are a few concerns that explanation is incomplete and deficient for the audience of Nature Communications to understand. It would be suitable for publication after adding and revising the following:

Major

Q1) This work appeals advanced droplet manipulations than the prior works which added magnetic additives into droplets or employed magnetically responsive array. Then, how about the moving velocity of the droplets? How fast can this system control the periodic magnetic field without delay? For example, the previous work employing magnetically bent array, S Jiang et al., *Small*, 14, 1702839 (2018), demonstrated quite fast droplet movements of about 2 cm/s.

Q2) Information about magnetic fields is lack. How large magnetic flux density is required to fold/unfold the JO-exosuit and to dispense/release daughter droplets? In addition, how can magnets array move toward demanded direction of droplet manipulations?

Q3) What is the novelty of this work, when compared to another capillary origami exhibiting self-folding, such as demonstration in M. F. Reynolds et al., *Nano Lett.* 19, 6221–6226 (2019)? The principle of the capillary origami should be articulated with the explanation and comparison of the previous works.

Q4) Why do you fabricate JO-exosuit through multiple steps, such as cutting and modifying the film by femtosecond laser after the film is prepared? How about just printing magnetic mixtures? As demonstrated in Y. Kim et al., *Nature*, 558, 274–279 (2018), I think the printing is simpler and more time/cost-effective, than methods using the femtosecond laser devices.

Q5) Why you align the magnetic particles along short side of the JO-exosuit for the tumbling manner? What happens the particles are aligned along the long side of the JO-exosuit or randomly dispersed? This should be described to emphasize the need to align magnetic particles for magnetic maneuver of JO-exosuit.

Q6) In this work, gradual folding of the JO-exosuit is generated by the surface tension of the liquid, such as water, NaOH, and HCL. How long does it take to fold, and should the folding be gradual? Does the magnitude of the surface tension affect the folding time? Is it possible to transport the droplet with large surface tension (e.g., liquid metal), such as demonstration in S. Jhang et al., *Adv. Mater. Interfaces*, 10, 2201247 (2022)?

Q7) In the case of stirring and mixing droplets in Figure 4g-i, I think it is impossible to remove JO-exosuit from the droplets without further wrapping liquids. However, in Figure 5 and page 9 line 387, it seems to transfer JO-exosuit from a liquid to another without folding. Does it happen with certain types of liquid, like buffer solution? What does “sliding out JO-exosuit using magnetic attraction” mean? Please kindly explain this mechanism in detail.

Q8) Sentences in this manuscript need to be rearranged and revised to improve readability. There is lack of connection between contents and emphasis on key principles and novelties. For example, it would be better to move the sentence in page 6 line 244, “The Janus characteristics of the JO-exosuit ~” to the head of the section.

Minor

Q1) page 3 line 102, “for directionally transport” should be changed to “to directionally transport” or “for directional transportation”.

Q2) page 3 line 120, It would be better to name them “top and bottom side”, than “upper and lower side”, when describing two different sides with Janus wettability.

Q3) page 5 line 177, Check the conjunctions between two sentences, “The modified surface is distributed with periodic bumped structures and covered with irregular micro-nanoparticles. While the unmodified surface is relatively flat.”.

Q4) page 6 line 232, Check typo in “interactions”.

Q5) page 8 line 325, There is no verb in sentence, “First, the JO-exosuit samples ~ larger droplet (~30 μ L), and then ~”.

Reply to the Reviewers' comments

Thanks a lot for organizing reviews of our manuscript (NCOMMS-23-09118A) and we sincerely appreciate all the reviewers for their positive evaluations and valuable comments. The reviewer's comments have been carefully considered and some additional experiments have been carried out. The manuscript has been carefully revised to address the reviewers' concerns. The major revisions were marked **in red** in the revised manuscript. The point-to-point answers to the comments are listed below.

Reviewer #1: The paper describes a multi-scale origami robot (exosuit) capable of cross-scale droplet manipulation with magnetic actuation. The results show that the origami robots perform various versatile droplet manipulation including three-dimensional transport, merging, splitting, releasing, stirring and remote heating of droplet of multiple size. The reviewer hopes the paper can be improved by addressing following comments.

- 1. In title, reviewer recommends replacing the name of the device (Exosuit) with “robot” or another word that can represent the device properly. “Exosuit” may mislead readers. It’s just recommendation. I understand why the authors name the device as an exosuit, but exosuit is for wearing not manipulating.**

Response: We thank the reviewer for this insightful suggestion. In order to properly represent the device, we replaced all the “exosuit” with “robot” in the revised manuscript and supplementary materials.

- 2. The authors show many functional capabilities of the proposed device. It would be a much more worthwhile paper, if the authors give a design principle for this droplet manipulation. For example, how to manage droplet volume in separation, relationship between carrying volume of droplet and design parameters of the device. If it is out of focus in this research, it would be helpful to provide discussion about main design parameters of the proposed device such as dimensional size, material treatment, shape of the device etc.**

Response: We greatly appreciate the reviewer for the constructive comment. We have added more discussion of design principles and design parameters to the revised manuscript and supplementary material (Figure R1-R10, Supplementary Figure 9, 16-22, 35-36).

The volume of the dispensed daughter droplet can be managed by controlling the dimensional size of the JO-robot. The relationship between the designed dimensional parameters of JO-robot and the volume of the dispensed and released droplets is illustrated in Figure R1. JO-robot can form a triangular prism after dispensing the droplet. The design volume of the dispensed droplet (V_d) can be approximately

considered as the volume of the triangular prism: $V_d = 0.5ac(b^2 - 0.25c^2)^{1/2}$, where a is the width of the JO-robot, b is the distance between the crease and the short edge, c is the distance between two creases (the creases are symmetrically distributed, Figure R1a). However, due to the surface tension of water (water surface protrudes from the triangular prism) and the bending deformation of the robot caused by the capillary force, the actual dispensed droplet volume is larger than the design volume (Figure R1c, Figure R2a). When the droplet is squeezed out, the design volume of the residual liquid (V_r) can be approximately considered as the volume of the JO-robot folding gap: $V_r = 0.5at(c-b)$, where t is the thickness of the JO-robot (Figure R1b). However, due to the elasticity of the JO-robot, the two plates cannot fit completely at the crease, resulting in residue at both creases, which makes the actual residue larger than the designed residue, that is to say, the volume of the actual squeezed droplet is slightly smaller than the design value (actual dispensed droplet volume minus design residual droplet volume, Figure R1d, Figure R2b).

Figure R1. Schematic diagram of the designed and the actual dispensed and residual liquid. (a, b) Relationship between the volume of dispensed/residual liquid and the size of JO-robot. (c, d) Optical images of actual dispensed droplet and residual liquid. Scale bars are 500 μm .

Figure R2. Comparison of experimental and design values of droplets that (a) dispensed and (b) released by JO-robots with different sizes.

In addition, more discussion on the design parameters that affect the droplet dispensing and release functions was added, such as the orientation of the magnetic particles chains, the number and distribution of creases and the shape of JO-robot.

As shown in Figure R3, when the magnetic particles are uniformly dispersed inside the JO-robot (without oriented magnetic particles chains), the JO-robot is not affected by the magnetic torque due to the lack of magnetic anisotropy, so it cannot be driven by the magnets array (magnets array moves to the right).

Figure R3. The JO-robot with uniformly dispersed magnetic particles cannot be driven by the magnets array. Scale bar is 5 mm. The inset is the enlarged view of JO-robot. Scale bar is 1mm.

When the carbonyl iron particles are aligned along the long side of the JO-robot (i.e., the angle between the chains and the long side is 0°), JO-robot (0°) tumbles around the short side to approach and touch the droplet (Figure R4). Since the tumbling axis is the short side, the JO-robot (0°) folds along the creases during the tumbling. Some of the liquid can be wrapped and dispensed by the JO-robot (0°) through the controlled tumbling. By increasing the magnetic field strength, the wrapped liquid can be squeezed out. **However, when the JO-robot (0°) tries to release the droplet by tumbling, the droplet is sucked back into the robot and cannot be released by the JO-robot (0°).** This is because the squeezed droplet is located on the side of the robot (relative to the tumbling direction, insets in Figure R4b) and it cannot be pushed away by the JO-robot with the superhydrophobic surface.

When the angle between the carbonyl iron particles chains and the long side is set to 30 degrees, the JO-robot (30°) folds along the creases and eventually tumbles around the short side as well (Figure R5). After contacting the water droplet, JO-robot (30°) can also achieve droplet dispensing by tumbling and wrapping part of the liquid. Similar to JO-robot (0°), **JO-robot (30°) is also difficult to release the dispensed daughter droplet due to the mismatch between the droplet location and the tumbling direction.**

Figure R4. Droplet manipulation by JO-robot with magnetic particles aligned along the long side (JO-robot (0°)). (a) Schematic diagram of JO-robot (0°). The dotted arrow indicates the orientation of the carbonyl iron particles chains. The right part shows the carbonyl iron particles chains inside the JO-robot (0°). Scale bar is $100\ \mu\text{m}$. (b) Droplet manipulation by JO-robot (0°). Droplet can be dispensed by JO-robot (0°), but cannot be released. The insets show the relative position of the squeezed droplet and the JO-robot. The squeezed droplet is located on the side of the JO-robot. Scale bar is $2\ \text{mm}$.

Figure R5. Droplet manipulation by JO-robot with an angle of 30 degrees between the magnetic particles chains and the long side (JO-robot (30°)). (a) Schematic diagram of JO-robot (30°). The dotted arrow indicates the orientation of the carbonyl iron particles chains. The right part shows the carbonyl iron particles chains inside the JO-robot (30°).

Scale bar is 100 μm . (b) Droplet manipulation by JO-robot (30°). Droplet can be dispensed by JO-robot (30°), but cannot be released. The insets show the relative position of the squeezed droplet and the JO-robot. The squeezed droplet is located on the side of the JO-robot. Scale bar is 2 mm.

When the angle between the carbonyl iron particles chains and the long side is set to 60 degrees, JO-robot (60°) tumbles around the long side. However, **the movement of JO-robot (60°) is very unstable**. The robot swings repeatedly during the movement. The unstable swing of the robot originates from the misalignment between the tumbling axis and the long side. The tumbling axis is perpendicular to the carbonyl iron particles chains, so the angle between the tumbling axis and the long side is 30 degrees. After wrapping the droplet, the JO-robot (60°) can also dispense the daughter droplet by tumbling. To release the droplet, the JO-robot (60°) turns the side carrying the droplet downward. When the magnetic field increases, the droplet between the robot and the superhydrophobic substrate can be squeezed out. After that, the JO-robot (60°) tumbles to detach the droplet for daughter droplet release (Figure R6). Although daughter droplet dispensing and release can be realized, **the controllability of the tumbling process and droplet manipulation process of the JO-robot (60°) is poor**.

The comparison experiments show that JO-robot can realize stable tumbling motion and droplet dispensing and release functions only when the carbonyl iron particles chains are designed to be perpendicular to the long side as demonstrated in the manuscript.

Figure R6. Droplet manipulation by JO-robot with an angle of 60 degrees between the magnetic particles chains and the long side (JO-robot (60°)). (a) Schematic diagram of JO-robot (60°). The dotted arrow indicates the orientation of the carbonyl iron particles chains. The right part shows the carbonyl iron particles chains inside the JO-robot (60°).

Scale bar is 100 μm . (b) Droplet manipulation by JO-robot (60°). Droplet can be dispensed and released by JO-robot (60°). Due to the misalignment between the tumbling axis and the long side, the controllability of the tumbling process and droplet manipulation process of the JO-robot (60°) is poor. Scale bar is 2 mm.

The effects of the number and distribution of creases, and the shape of JO-robot on the droplet manipulation performance are discussed in Figures R7-R10. Here, the rectangular sheet-like structure is chosen for the design of JO-robot due to its extremely simple design, ease to manufacture and capability of various motion modes. The rectangular sheet-like soft structure has been recognized as an effective design and widely used in the field of magnetic micro-robots (*Nature* **554**, 81–85 (2018); *Sci. Adv.* **7**, eabh2022 (2021)). We first study the effect of crease number on droplet manipulation. As shown in Figure R7a, the JO-robots with 0 to 4 creases are used for daughter droplet dispensing and release, respectively. The JO-robots with different number of creases can dispense daughter droplet by folding along the creases. By increasing the magnetic field strength, JO-robots can squeeze out daughter droplets and release the droplets by tumbling and pushing (Figure R7b-f). Among them, it is worth mentioning that the dispensing and release processes are slightly different for the JO-robot with no crease (NC-JO-robot). When dispensing daughter droplet, the NC-JO-robot bends along the central axis to wrap the liquid and detaches it by tumbling. When NC-JO-robot folds to release the daughter droplet, due to the elasticity of the robot, there is always a gap in the middle part of the folded NC-JO-robot. The droplet is difficult to be completely squeezed out, resulting in the failure of the release. In order to release the daughter droplet, the NC-JO-robot is first unfolded by the magnetic field and then squeezes out the daughter droplet by interaction with the substrate. The NC-JO-robot can eventually realize daughter droplet release by using the superhydrophobic outer surface to gently push the droplet. The volume of daughter droplets that can be dispensed and released by JO-robots with different number of creases are shown in Figure R7g, h, respectively. It shows that the JO-robots with 1 and 3 creases dispense slightly smaller droplets than other robots. The NC-JO-robot can dispense the largest droplet, but its operation is complicated as described above. Therefore, the JO-robot with two creases can meet the requirements of processing efficiency, manipulation convenience and droplet manipulation performance.

The droplet manipulation by JO-robots with different crease distributions is shown in Figure R8. The total length of the JO-robots remains the same (7 mm), while the distance between the crease and the short side changes from 1 mm to 3 mm (Figure R8a). When the middle plate is longer than the side plates (JO-robot can be seen as three plates connected by two creases), the plates on both sides can fold symmetrically toward the middle plate when dispensing droplets (Figure R8b-d). When the crease distances are 1 mm and 1.5 mm, the JO-robots cannot fold to form the closed triangles. However, due to the origami deformation ability and the Janus wetting characteristic, they can still dispense and release droplets (Figure R8b, c). When the length of the two side plates is longer than the middle plate, the JO-robot tends to fold asymmetrically

along the side plates when dispensing droplets, making the dispensing process less controllable (Figure R8e, f). The volume of daughter droplets that can be dispensed and released by JO-robots with different crease distributions are shown in Figure R8g, h, respectively. In conclusion, when the lengths of the three plates of the JO-robot conform to the dimensional relationship forming a triangle and the lengths of the side plates are shorter than the length of the middle plate, the JO-robot can fold symmetrically along the middle microplate to achieve stable droplet manipulation (for example the length of the side plate is 2 mm and the length of the middle plate is 3 mm as demonstrated in the manuscript).

Figure R7. Daughter droplet dispensing and release by JO-robots with different number of creases. (a) The schematic diagram of the JO-robots with different number of creases. (b-f) JO-robots with 0 to 4 creases are used for daughter droplet dispensing and release, respectively. Scale bar is 5 mm. (g, h) Volume of daughter droplets that can be (g) dispensed and (h) released by JO-robots with different creases. The insets show the top view of the JO-robots wrapping water droplets.

Figure R8. Daughter droplet dispensing and release by JO-robots with different crease distributions. (a) The schematic diagram of the JO-robots with different crease distributions. (b-f) JO-robots with the distance between the crease and the short side from 1 mm to 3 mm are used for daughter droplet dispensing and release, respectively. Scale bar is 5 mm. (g, h) Volume of daughter droplets that can be (f) dispensed and (g) released by JO-robots with different crease distances. The insets show the top view of the JO-robots wrapping water droplets.

The droplet manipulation by JO-robots with different widths is shown in Figure R9. JO-robots with widths from 1 mm to 4 mm are used for daughter droplet dispensing and release, respectively (Figure R9a-e). The JO-robots with different widths can realize the droplet dispensing and release. The droplet volume that can be dispensed and released by the JO-robots with different widths are shown in Figure R9f, g, respectively. The volume of the dispensed and released droplets increases with the width of the JO-robot. However, as discussed in Figure R1, the residual liquid also increases with the width of the JO-robot.

The droplet manipulation by JO-robots with different lengths is shown in Figure R10. The lengths of the middle plates are kept at 3 mm, while the lengths of the side plates are 1 mm to 4 mm, respectively (Figure R10a). The JO-robots with side plate lengths ranging from 1 mm to 4 mm can dispense and release daughter droplets (Figure R10b-e). When the length of the side plate is 1 mm, the JO-robot cannot fold to form a closed triangle (Figure R10b). As mentioned before, the JO-robot folds asymmetrically along the side plates when the middle plate is shorter than the side plates, making the dispensing process less controllable (Figure R10e). The droplet volume that can be dispensed and released by the JO-robots with different lengths are shown in Figure R10f, g, respectively. The volume of the dispensed and released droplets increases with the length of the JO-robot.

In summary, JO-robots with different design parameters can realize the droplet dispensing and release functions by virtue of origami deformation ability and the Janus wetting characteristic. Considering the factors such as droplet manipulation performance, manipulation convenience and processing efficiency, the design criteria of JO-robot include the following three aspects:

- 1) JO-robot is composed of three plates (i.e., with two creases);**
- 2) The lengths of the three plates can conform to a closed triangle;**
- 3) The length of the middle plate is longer than that of the side plate.**

Figure R9. Daughter droplet dispensing and release by JO-robots with different widths. (a) The schematic diagram of the JO-robots with different widths. (b-e) JO-robots with widths from 1 mm to 4 mm are used for daughter droplet dispensing and release, respectively. Scale bar is 5 mm. (f, g) Volume of daughter droplets that can be (f) dispensed and (g) released by JO-robots with different widths.

Figure R10. Daughter droplet dispensing and release by JO-robots with different lengths. (a) The schematic diagram of the JO-robots with different lengths. (b-e) JO-robots with lengths from 5 mm to 11 mm are used for daughter droplet dispensing and release, respectively. Scale bar is 5 mm. (f, g) Volume of daughter droplets that can be (f) dispensed and (g) released by JO-robots with different widths. The insets show the top view of the JO-robots wrapping water droplets.

Revision: Following the reviewer’s constructive comments, we added the discussion of design principles and design parameters to the revised manuscript and supplementary material (Supplementary Discussion 3, Supplementary Figure 9, 16-22, 35-36). The revisions to the original manuscript are marked **in red**.

Page 5, Line 192 – Line 198: *When the chains are distributed perpendicularly to the long side of the robot, the tumbling axis overlaps with the long side. Therefore, the JO-robot can stably tumble around the long side to approach and manipulate the droplet under the action of the moving periodic magnetic field generated by magnets array (Supplementary Fig. 7, 8). In contrast, when the carbonyl iron particles are uniformly distributed, the robot cannot be driven by the magnetic field (Supplementary Fig. 9).*

Page 7, Line 289 – Line 291: *The relationship between the designed dimensional parameters and the volume of the dispensed and released droplets is shown in Supplementary Discussion 3.*

Page 8, Line 309 – Line 312: *In addition, we further discuss the effects of different design parameters on droplet dispensing and release, such as the orientation of carbonyl iron particles chains, the number and distribution of creases, and the width and length of the JO-robot (Supplementary Fig.16-22).*

Additional Figures (*Supplementary Figure 9, 16-22, 35-36*), *corresponding captions* and *Supplementary Discussion 3* have been added to the revised supplementary material.

- 3. Except for the functional performance of the device, could you summarize enabling technologies or key design factors for manipulating droplet with magnetic origami? It is helpful to understand why exiting technology couldn't realize these functions. Please compare with following paper.**
Tianqi Xu et al. “Millimeter-scale flexible robots with programmable three-dimensional magnetization and motions”, Science robotics, 2019

Response: We thank the reviewer for this helpful suggestion. We believe that the Janus wetting characteristic endows the JO-robot with unique versatile droplet manipulation abilities when compared with the planar microrobots prepared by ultraviolet (UV) lithography (*Sci. Robot.* **4**, eaav4494 (2019)). And compared with the existing magnetic actuation droplet manipulation technologies, the combination of origami deformation ability and the Janus wetting characteristic gives the JO-robot unique droplet manipulation ability.

The preparation method based on UV lithography has distinct advantages in the preparation of planar composite microrobots due to its high preparation precision, one-process fabrication and high magnetization encodability properties. This method allows the preparation of diversified planar microrobots with distributed three-dimensional magnetization profiles. Compared to the robots shown in the literature (*Sci. Robot.* **4**, eaav4494 (2019)), JO-robot is also a kind of planar composite microrobot, and the difference is the rational design of the Janus wetting characteristic (i.e., two surfaces have entirely different wetting properties). The Janus wetting characteristic gives the JO-robot unique advantages in versatile droplet manipulation, especially in the application of daughter droplet dispensing and release:

(1) Daughter droplet dispensing.

As shown in Figure R11a, the JO-robot can realize controllable droplet dispensing. The laser modified top surface of the JO-robot is superhydrophobic with low droplet adhesion and the unmodified bottom side is hydrophobic with high droplet adhesion. The hydrophobic surface enables JO-robot to firmly wrap the liquid during the dispensing process. At the same time, the superhydrophobic surface ensures that the outer surface of the JO-robot will not be wetted by the liquid, thus facilitating the formation and breakage of the necking point, which is the key to the precise droplet dispensing. In comparison, the liquid wets the outer surface of the non-Janus origami robot (both two sides are hydrophobic with high droplet adhesion) and gradually slides on the surface as the robot tumbles, eventually forming a residue on the robot when it disengages (Figure R11b).

Figure R11. Comparison of the daughter droplet dispensing process between JO-robot and non-Janus origami robot. (a) Controllable daughter droplet dispensing by JO-robot. (b) Daughter droplet dispensing by non-Janus origami robot. Scale bars is 2 mm.

(2) Daughter droplet release.

As shown in Figure R12a, the JO-robot first squeezes out the wrapped droplet by folding deformation. And then, the JO-robot uses its superhydrophobic outer surface to gently push the droplet to separate the droplet from the robot. Finally, the JO-robot tumbles in reverse to achieve on-demand droplet release. In contrast, when the non-Janus origami robot folds to squeeze out the wrapped droplet, the plates on both sides of the robot interfere with each other to form a ridge due to the same wettability of the inner and outer surfaces. Some liquid remains in this ridge and is difficult to be completely squeezed out. Furthermore, since the outer surface of the non-Janus origami robot is also hydrophobic and has a high droplet adhesion, the droplet always sticks to the outer surface of the robot and cannot be released by the non-Janus origami robot (Figure R12b).

Figure R12. Comparison of the droplet release process between JO-robot and non-Janus origami robot. (a) On-demand droplet release by JO-robot. (b) Droplet release process by the non-Janus origami robot. The non-Janus origami robot cannot release the droplet due to the high adhesion between droplet and the outer surface of the robot. Scale bars is 2 mm.

In summary, the rational design of Janus wetting characteristic endows the JO-robot with unique droplet manipulation ability. It enables JO-robot to realize controllable daughter droplet dispensing and release, which are difficult to be realized by other planar magnetic microrobots without Janus wetting characteristic. By the way, this Janus wetting characteristic can be obtained without additional surface coating. The surface wettability modification (surface roughening), creation of the robot outer contour and creases can be achieved through a single femtosecond laser processing step. In addition, even at small scale and at the water-oil interface, the Janus wetting characteristic is still effective, so JO-robot can also realize multifunctional droplet manipulation at small scale (Figure 4i) and in oil environment (Figure 5d).

In addition, to summarize the key design factors more clearly, we compared the JO-robot-based droplet manipulation strategy with the existing magnetic actuation droplet manipulation strategies as the reviewer suggested.

The current magnetic actuation droplet manipulation strategies can be generally classified into two main categories (Table R1, R2 and Supplementary Table 1, 2): (1) adding magnetic additives into droplets and (2) adding magnetically responsive materials into the elastomeric substrate surfaces.

(1) Magnetic additives-based droplet manipulation strategies.

Compared with the magnetic additives-based droplet manipulation strategies, the combination of origami deformation ability and the Janus wetting characteristic gives the JO-robot unique daughter droplet release ability.

As shown in the Table R1, magnetic additives can dispense daughter droplet from the large droplet. However, due to the lack of interaction between magnetic additives and dispensed daughter droplet, the daughter droplet remains entangled with the additives and is difficult to be released on-demand. In order to address the issue of droplet contamination, additional purification steps are required to acquire pure daughter droplet (*Sci. Adv.* **6**, eaay5808 (2020); *Nat. Commun.* **12**, 7136 (2021)). In the other case, the daughter droplets are fixed in the design area (*Adv. Mater.* **25**, 2903-2908 (2013)) or are randomly scattered (*Sci. Adv.* **7**, eabi7498 (2021)). As mentioned before, the JO-robot can interact with the daughter droplet through folding deformation (or origami deformation) and pushing motion. It can squeeze out the wrapped droplet through origami deformation and use the superhydrophobic outer surface to gently push the droplet to realize controllable release.

Table R1. Comparison of droplet manipulation functions that can be achieved by magnetic additives-based droplet manipulation strategies.

 Magnetic additive Function	 Hydrophobic magnetic powder (Adv. Funct. Mater. 25, 437-444 (2015))	 Hydrophilic magnetic particles (Adv. Mater. 25, 2903-2908 (2013))	 Steel beads (Sci. Adv. 6, eaay5808 (2020))
Daughter droplet dispensing/Splitting	✘	✔	✔
On-demand daughter droplet release	✘	✘	✘
 Magnetic additive Function	 Printed solid frame (Sci. Adv. 7, eabi7498 (2021))	 Ferrofluid (Nat. Commun. 12, 7136 (2021).)	 JO-robot (This work)
Daughter droplet dispensing/Splitting	✔	✔	✔
On-demand daughter droplet release	✘	✘	✔

* Most magnetic additives-based droplet manipulation strategies can realize basic functions of droplet manipulation such as transportation, merging and mixing, so they are not discussed here.

(2) Magneto-responsive surfaces-based droplet manipulation strategies.

Compared with the magneto-responsive surfaces-based droplet manipulation strategies, the untethered moving ability, origami deformation ability and the Janus wetting characteristic give JO-robot versatile droplet manipulation ability.

Due to the fixed substrates and simple structure deformation capabilities (such as bending or depression), the magneto-responsive surfaces-based droplet manipulation strategies are only capable of simple droplet manipulations such as transport, merging and mixing. The daughter droplet dispensing and release have not been realized by these strategies (Table R2).

Table R2. Comparison of droplet manipulation functions that can be achieved by magneto-responsive surfaces-based droplet manipulation strategies.

Substrate Function	 Microcolumns (Adv. Funct. Mater. 28 , 1800163 (2018).)	 Microdimple (ACS Appl. Mater. Interfaces 11 , 47642-47648 (2019))	 Microcilia (Proc. Natl. Acad. Sci. U.S.A. 118 , e2111291118 (2021))
	Daughter droplet dispensing/Splitting	✗	✗
On-demand daughter droplet release	✗	✗	✗
Substrate Function	 Microplate (Nano Lett. 20 , 7519-7529 (2020))	 Microtube (Sci. Adv. 4 , eaau8767 (2018))	 JO-robot (This work)
	Daughter droplet dispensing/Splitting	✗	✗
On-demand daughter droplet release	✗	✗	✓

* Most magneto-responsive surfaces-based droplet manipulation strategies can realize basic functions of droplet manipulation such as transportation, merging and mixing, so they are not discussed here.

In conclusion, compared with the planar magnetic microrobots prepared by UV lithography, the rational design of Janus wetting characteristic endows the JO-robot with controllable daughter droplet dispensing and on-demand daughter droplet release abilities. Compared with the existing magnetic actuation droplet manipulation technologies, the combination of origami deformation ability and the Janus wetting characteristic gives JO-robot unique versatile droplet manipulation ability. Based on these unique features, JO-robot can realize on-demand generation of microdroplets array, which is incapable with other magnetic actuation droplet manipulation strategies (Fig. 3g). Furthermore, these key design factors are still effective at small scale and at the water-oil interface, so the JO-robot can realize versatile cross-scale droplet manipulation (Figure 4i) and can also manipulate water droplets in oil environment (Figure 5d).

Revision: We summarized the key design factors in the revised manuscript following the reviewers' suggestions. The corresponding reference has been added as Ref. 30.

Page 3, Line 96 – Line 98: *Compared with the existing magnetic actuation droplet manipulation methods, the combination of origami deformation ability and Janus wetting characteristic endows JO-robot with unique droplet manipulation abilities.*

Page 4, Line 170 – Line 173: *With the advantage of efficient maskless processing, the cutting of JO-robot contour, the creation of creases and surface modification of the top side can be achieved in one step by the femtosecond laser processing, which is difficult to achieve by other magnetic microrobots fabrication methods^{29,30}.*

Page 6, Line 222 – Line 224: *The combination of origami deformation ability and the Janus wetting characteristic endows JO-robot with controllable daughter droplet dispensing and release abilities.*

Page 6, Line 246 – Line 247: *With the origami deformation ability and the Janus wetting characteristic, JO-robot can realize on-demand daughter droplet release.*

Page 6, Line 255 – Page 7, Line 264: *During the dispensing and release processes, the Janus wetting characteristic of JO-robot plays the key role. The superhydrophobic outer surface is not wetted by droplet, which promotes the formation of the necking point, facilitates the dispensing of daughter droplet and avoids unnecessary liquid residue. It also ensures that the droplet does not stick to the JO-robot during release. Although other preparation methods such as 3D printing and ultraviolet lithography can easily prepare magnetic robots, they need additional surface modification to obtain Janus wetting characteristic^{29,30}. The comparison tests of the droplet dispensing and release of magnetic robot without the Janus characteristic are shown in Supplementary Fig. 13, 14.*

Page 12, Line 506 – Line 511: *Compared with other magnetic actuation droplet manipulation strategies, the combination of origami deformation ability and Janus wetting characteristic endows JO-robot with higher flexibility and wider droplet volume applicability, while avoiding the mixing of droplets with magnetic additives, simplifying the droplet manipulation steps and avoiding the risk of droplet contamination.*

Finally, we thank the reviewer again for these thoughtful comments. The manuscript has greatly benefited from these insightful suggestions.

Reviewer #2: The manuscript by Jiang et al. reports on the three-dimensional manipulation of droplets utilizing millimeter scale origami-based robot assisted by conjunction of magnetic force and capillary force. Authors nicely demonstrated dispensing, releasing, stirring, and mixing droplets by designing robots with aligned magnetic particles and Janus wettability of superhydrophobic and hydrophobic properties. Especially, extraction and purification of nucleic acids is an outstanding demonstration by untethered stirring and dispensing of the droplets. However, there are a few concerns that explanation is incomplete and deficient for the audience of Nature Communications to understand. It would be suitable for publication after adding and revising the following:

- 1. This work appeals advanced droplet manipulations than the prior works which added magnetic additives into droplets or employed magnetically responsive array. Then, how about the moving velocity of the droplets? How fast can this system control the periodic magnetic field without delay? For example, the previous work employing magnetically bent array, S Jiang et al., *Small*, **14**, 1702839 (2018), demonstrated quite fast droplet movements of about 2 cm/s.**

Response: We are grateful for the very positive comments from the reviewer. We have tested the droplet transport speed as shown in Figure R13-R15. It should be noted here that, as suggested by one of the reviewers, we replaced the “exosuit” (JO-exosuit) with “robot” (JO-robot) in the revised manuscript to properly represent the device.

As the reviewer mentioned, the droplet self-propelling induced by Laplace pressure difference can realize fast movement with a speed of 2 cm/s (*Small*, **14**, 1702839 (2018)). Unlike droplet self-propelling, the JO-robot can actively transport water droplet through its untethered motion (Figure R13). As shown in Figure R14, the JO-robot is driven by the horizontal movement of the magnets array. The periodic magnetic field is generated by the magnets array. When the magnets array is controlled by the drive system, the periodic magnetic field moves together with the magnets array without control delay. The droplet transport speed increases with the moving speed of the magnets array and it can reach 68.47 ± 2.1 mm/s (6.5 μ L droplet, corresponding to the magnets array moving speed of 300 mm/s), which is much faster than the droplet self-propelling speed (~ 20 mm/s, *Small*, **14**, 1702839 (2018)).

It should be mentioned that the higher magnets array moving speed may cause the droplet to be thrown out of the robot before the transportation delay occurs, which makes the transport ineffective. For example, when the moving speed of the magnets array reaches 310 mm/s, the JO-robot can still tumble forward at a speed of 72.24 ± 1.82 mm/s. However, some of the liquid wrapped in the JO-robot is thrown out during the high-speed transportation, resulting in uncontrollable leakage (Figure R15). As the droplet volume decreases, the droplet transport speed can be further increased. For

example, the transport speed of a 4 μL droplet can reach 78.84 ± 3.4 mm/s with the magnets array moving speed of 360 mm/s.

Figure R13. High-speed droplet transportation by JO-robot (transport speed is about 69.9 mm/s). Scale bar is 5 mm.

Figure R14. Droplet transport speed versus magnets array moving speed. The inset shows the drive system of the magnets array.

Figure R15. Liquid is thrown out during high-speed transportation above 72.24 ± 1.82 mm/s. Scale bar is 5 mm.

Revision: According to the reviewer’s comments, we have added the discussion of the droplet transport speed to the revised manuscript and supplementary material (Supplementary Figure 1). The corresponding reference has been added as Ref. 27. The revisions to the original manuscript are marked **in red**.

Laplace pressure difference²⁷, the droplet transport speed of JO-robot can reach as high as 68.47 ± 2.1 mm/s (6.5 μ L droplet, Supplementary Fig. 1).

Additional Figure (*Supplementary Fig. 1*) and *corresponding caption* have been added to the revised supplementary material.

- 2. Information about magnetic fields is lack. How large magnetic flux density is required to fold/unfold the JO-exosuit and to dispense/release daughter droplets? In addition, how can magnets array move toward demanded direction of droplet manipulations?**

Response: We thank the reviewer for the comment. According to our experiments, the magnetic flux density required for the tumbling and folding of the JO-robot, the dispensing and transportation of droplet is between 81.5 ± 5.7 mT and 155.4 ± 10.4 mT. The release of droplet by JO-robot requires the magnetic flux density larger than 265.0 ± 24.7 mT.

When **the magnetic flux density is larger than 16.9 ± 1.8 mT**, JO-robot (without carrying the droplet) can tumble forward (Figure R16a).

The magnetic flux density required for the droplet dispensing by JO-robot is between 81.5 ± 5.7 mT and 155.4 ± 10.4 mT. When the magnetic flux density is smaller than 81.5 ± 5.7 mT, the JO-robot can be pulled back to the initial position by the capillary force during the tumbling process, which leads to the failure of dispensing (Figure R16b). When the magnetic flux density is larger than 155.4 ± 10.4 mT, the JO-robot subjects to a greater magnetic force than the capillary force. During the dispensing process, the plates on both sides of the JO-robot (the JO-robot can be seen as three plates connected by two creases) can be unfolded. A portion of the liquid can adhere to the hydrophobic surface (with high droplet adhesion) of the JO-robot and be dispensed. Although JO-robot can dispense the droplet, it cannot transport the droplet under this magnetic flux density. When JO-robot tumbles, it interacts with the substrate and squeeze out the droplet, resulting in the failure of transportation (Figure R16c).

The magnetic flux density required for the droplet transportation is between 29.0 ± 2.6 mT and 227.4 ± 13.9 mT. When the magnetic flux density is smaller than 29.0 ± 2.6 mT, the droplet is difficult to be transported by the JO-robot, while when the magnetic flux density is larger than 227.4 ± 13.9 mT, the droplet can be squeezed out by the robot during transportation, resulting in the failure of transportation (Figure R16d).

In summary, in order to enable tumbling motion, folding, dispensing, and transporting of droplet (these processes are usually composed as a continuous droplet manipulation target), the magnetic flux density needs to be controlled between 81.5 ± 5.7 mT and 155.4 ± 10.4 mT.

In order to release the droplet, the magnetic flux density should be larger than 265.0 ± 24.7 mT, so that the JO-robot can squeeze out and release the droplet (Figure R16e).

Figure R16. Droplet manipulation by JO-robot under different magnetic flux densities. Scale bar is 2 mm.

The schematic diagram of the magnetic drive system is shown in Figure R17. The distance between the magnets array and JO-robot can be adjusted by the Z-axis lifting platform to control the magnetic flux density. The magnets array is carried by the sliders and moves horizontally on the X-axis guide rail to drive the JO-robot for droplet manipulation.

Figure R17. Schematic diagram of the magnetic drive system.

Revision: To respond to the reviewer's comments, the magnetic field information and the schematic diagram of the magnetic drive system have been added to the revised manuscript and supplementary material (Supplementary Figure 7).

Page 6, Line 252 – Line 255: *In brief, on-demand daughter droplet release is achieved by deliberate squeezing and push-off motions (with the magnetic flux density larger*

than 265.0 ± 24.7 mT, Supplementary Movie 6).

Page 8, Line 323 – Line 325: *The magnetic flux density required for droplet manipulation including the tumbling of JO-robot, the dispensing and transportation of droplet is between 81.5 ± 5.7 mT and 155.4 ± 10.4 mT.*

Additional Figure (*Supplementary Fig. 7*) and *corresponding caption* have been added to the revised supplementary material.

3. What is the novelty of this work, when compared to another capillary origami exhibiting self-folding, such as demonstration in M. F. Reynolds et al., *Nano Lett.* **19, 6221–6226 (2019)? The principle of the capillary origami should be articulated with the explanation and comparison of the previous works.**

Response: We thank the reviewer for the comment. Capillary origami is a planar device capable of spontaneous wrapping of the droplet by the elastic substrate. The droplet on the elastic substrate tends to reduce the liquid-air interface, thus lowering the surface energy at the cost of increasing the elastic potential energy of the elastic substrate (*Phys. Rev. Lett.* **98**, 156103 (2007); *Appl. Phys. Lett.* **97**, 014103 (2010)). Three-dimensional (3D) structures can be constructed by the interaction of the elasticity and capillarity.

The previous works mainly focused on the capillary origami of nonresponsive substrates such as molybdenum disulfide (MoS₂), polydimethylsiloxane (PDMS) and polystyrene (PS), etc. (*Phys. Rev. Lett.* **98**, 156103 (2007); *Nano Lett.* **19**, 6221-6226 (2019); *Appl. Phys. Lett.* **97**, 014103 (2010); *Nat. Mater.*, **14**, 1206 (2015), etc.). For example, Michael F. Reynolds et al. use the droplet to fold atomically thin two-dimensional materials (MoS₂) and planar microstructures connected by MoS₂ hinges (*Nano Lett.* **19**, 6221-6226 (2019)). Based on the capillary origami, they have demonstrated a variety of exquisite 3D structures. However, due to the limitation of the material properties, the assembled 3D structures lack functionality, so the authors mentioned in their outlook: “*The addition of active panels that include electronic, optical, or magnetic materials could permit the design of three-dimensional structures with functional properties.*”

In this work, the magnetic substrate with the rational design of oriented magnetic particles chains and Janus wetting characteristic (JO-robot) gives the origami structure more functionality. We added magnetically responsive particles to the elastic substrate and intentionally align them into chains to endow the elastic substrate with untethered moving ability and magnetic responsive deformability. The three-dimensional structure constructed by the capillary origami can be further manipulated by the magnetic field. The combination of magnetic responsive performance and Janus wetting characteristic allows the elastic substrate (JO-robot) to realize versatile droplet manipulation through capillary origami.

Revision: According to the reviewer's comments, we have added more discussion of the principle of the capillary origami and the comparison of the previous works to the revised manuscript. The corresponding reference has been added as Ref. 26.

Page 4, Line 133 – Line 144: *Two shallow creases are fabricated on the top surface. JO-robot tumbles directionally under the drive of the magnetic field and it can wrap the droplet with the assistance of capillary force and magnetic field. When the JO-robot comes into contact with the droplet, the droplet tends to reduce the liquid-air interface, thus lowering the surface energy. The capillary force generated in this process deforms the structure, thus increasing the elastic potential energy of the JO-robot, causing the robot to fold along the creases²⁵. The magnetically controlled motion of the JO-robot allows both creases to contact the droplet and thus be folded. Compared with previous works that mainly focused on the capillary origami of nonresponsive substrates, such as molybdenum disulfide, polydimethylsiloxane, etc.^{25,26}, here, the magnetic substrate with the rational design of oriented magnetic particles chains and Janus wetting characteristic gives the origami structure diverse functionalities.*

4. Why do you fabricate JO-exosuit through multiple steps, such as cutting and modifying the film by femtosecond laser after the film is prepared? How about just printing magnetic mixtures? As demonstrated in Y. Kim et al., Nature, 558, 274–279 (2018), I think the printing is simpler and more time/cost-effective, than methods using the femtosecond laser devices.

Response: We thank the reviewer for raising such a good question. After the magnetic film is prepared, it requires precise laser cutting and laser modification. The purpose of laser cutting is to obtain samples of specific size, shape and creases, while the purpose of laser modification is to endow the JO-robot with Janus wetting characteristic (the laser modified top surface is superhydrophobic with low droplet adhesion and the unmodified bottom side is hydrophobic with high droplet adhesion).

The Janus wetting characteristic is the key for JO-robot to realize versatile droplet manipulation, especially for daughter droplet dispensing and release. Therefore, after preparing the magnetic film, it is necessary to modify the surface to obtain the Janus wetting characteristic. The comparison experiments of the daughter droplet dispensing and release between JO-robot and non-Janus robot are shown in the Figure R18, R19.

(1) Daughter droplet dispensing.

As shown in Figure R18a, the JO-robot can realize controllable droplet dispensing. The laser modified top surface of the JO-robot is superhydrophobic with low droplet adhesion and the unmodified bottom side is hydrophobic with high droplet adhesion. The hydrophobic surface enables JO-robot to firmly wrap the liquid during the

dispensing process. At the same time, the superhydrophobic surface ensures that the outer surface of the JO-robot will not be wetted by the liquid, thus facilitating the formation and breakage of the necking point, which is the key to the precise droplet dispensing. In comparison, the liquid wets the outer surface of the non-Janus origami robot (both two sides are hydrophobic with high droplet adhesion) and gradually slides on the surface as the robot tumbles, eventually forming a residue on the robot when it disengages (Figure R18b).

Figure R18. Comparison of the daughter droplet dispensing process between JO-robot and non-Janus origami robot. (a) Controllable daughter droplet dispensing by JO-robot. (b) Daughter droplet dispensing by non-Janus origami robot. Scale bars is 2 mm.

(2) Daughter droplet release.

As shown in Figure R19a, the JO-robot first squeezes out the wrapped droplet by folding deformation. And then, the JO-robot uses its superhydrophobic outer surface to gently push the droplet to separate the droplet from the robot. Finally, the JO-robot tumbles in reverse to achieve on-demand droplet release. In contrast, when the non-Janus origami robot folds to squeeze out the wrapped droplet, the plates on both sides of the robot interfere with each other to form a ridge due to the same wettability of the inner and outer surfaces. Some liquid remains in this ridge and is difficult to be completely squeezed out. Furthermore, since the outer surface of the

non-Janus origami robot is also hydrophobic and has a high droplet adhesion, the droplet always sticks to the outer surface of the robot and cannot be released by the non-Janus origami robot (Figure R19b).

Figure R19. Comparison of the droplet release process between JO-robot and non-Janus origami robot. (a) On-demand droplet release by JO-robot. (b) Droplet release process by the non-Janus origami robot. The non-Janus origami robot cannot release the droplet due to the high adhesion between droplet and the outer surface of the robot. Scale bars is 2 mm.

As can be seen from the comparison experiments, the Janus wetting characteristic enables JO-robot to realize controllable daughter droplet dispensing and release, which is difficult to achieve by non-Janus robot. So, laser cutting and surface modification of the magnetic film are necessary for JO-robot to realize versatile droplet manipulation. In fact, the cutting and modification are achieved in a single laser processing step. The contour cutting path (including the crease processing path) and surface wettability modification path are shown in Figure R20. After the laser scans the outer contour and creases of the JO-robot along the designed path (scanning path 1), it quickly scans the

top surface along the grid path (scanning path 2) to achieve the modification of the wettability (roughening of the top surface). The contour cutting, creases scanning and wettability modification of the JO-robot can be achieved in a single processing step by importing the designed scanning paths into the software, setting the path processing sequence and processing speed.

Figure R20. Schematic diagram of the laser scanning paths during processing.

Now, we come back to the direct printing method. 3D printing is an efficient method for the preparation of planar or 3D structures. The printing method that mentioned in the literature (*Nature*, **558**, 274-279 (2018)) enables direct ink writing for the rapid preparation of delicate magnetic robots. However, the planar robots prepared by the 3D printing method require additional surface modification to obtain the essential Janus wetting characteristic. Femtosecond laser processing can achieve contour cutting, crease scanning and surface modification through one-step processing, even for the small-sized JO-robot ($230\ \mu\text{m} \times 940\ \mu\text{m}$). So, we believe that for the processing of JO-robot, femtosecond laser processing method has unique advantageous compared with 3D printing.

Revision: Following the reviewer’s comments, we added more discussion to the revised manuscript. The corresponding reference has been added as Ref. 29.

Page 4, Line 170 – Line 173: *With the advantage of efficient maskless processing, the cutting of JO-robot contour, the creation of creases and surface modification of the top side can be achieved in one step by the femtosecond laser processing, which is difficult to achieve by other magnetic microrobots fabrication methods^{29,30}.*

Page 6, Line 259 – Page 7, Line 262: *Although other preparation methods such as 3D printing and ultraviolet lithography can easily prepare magnetic robots, they need additional surface modification to obtain Janus wetting characteristic^{29,30}.*

Page 14, Line 574 – Line 580: *The outer contour (300 mW, 10 mm/s, 14 circles) and creases (300 mW, 50 mm/s, 14 circles) of the JO-robot and the modification of the top surface (300 mW, 50 mm/s, 3 circles, grid scanning at 60 μm intervals) are processed in one step by laser scanning.*

5. Why you align the magnetic particles along short side of the JO-exosuit for the tumbling manner? What happens the particles are aligned along the long side of the JO-exosuit or randomly dispersed? This should be described to emphasize the need to align magnetic particles for magnetic maneuver of JO-exosuit.

Response: We thank the reviewer for the valuable comments. The magnetic particles are aligned into chains to create magnetic anisotropy. When the magnetic field is applied to the chains, magnetic torques can be generated within the chains to align them with the direction of the magnetic field (*Nat. Mater.* **10**, 747–752 (2011); *ACS Appl. Mater. Int.* **9**, 11895–11901 (2017)). Therefore, the magnetic torques always drive the JO-robot to tumble around the direction perpendicular to the iron particles chains (i.e., the tumbling axis is perpendicular to the direction of the chains). When the magnetic particles chains are aligned along the short side of the JO-robot, the tumbling axis overlaps with the long side. The JO-robot (without wrapping droplet) can tumble stably around the long side and realize controllable droplet manipulation functions. To respond to the reviewer’s suggestions, we conducted a series of comparison experiments on JO-robot with different magnetic particles chains orientations (Figure R21-R24).

As shown in Figure R21, when the magnetic particles are randomly dispersed inside the JO-robot (without oriented magnetic particles chains), the JO-robot is not affected by the magnetic torque due to the lack of magnetic anisotropy, so it cannot be driven by the magnets array (magnets array moves to the right).

Figure R21. The JO-robot with randomly dispersed magnetic particles cannot be driven by the magnets array. Scale bar is 5 mm. The inset is the enlarged view of JO-robot. Scale bar is 1mm.

When the carbonyl iron particles are aligned along the long side of the JO-robot (i.e., the angle between the chains and the long side is 0°), JO-robot (0°) tumbles around the

short side to approach and touch the droplet (Figure R22). Since the tumbling axis is the short side, the JO-robot (0°) folds along the creases during the tumbling. Some of the liquid can be wrapped and dispensed by the JO-robot (0°) through the controlled tumbling. By increasing the magnetic field strength, the wrapped liquid can be squeezed out. **However, when the JO-robot (0°) tries to release the droplet by tumbling, the droplet is sucked back into the robot and cannot be released by the JO-robot (0°).** This is because the squeezed droplet is located on the side of the robot (relative to the tumbling direction, insets in Figure R22b) and it cannot be pushed away by the JO-robot with the superhydrophobic surface.

Figure R22. Droplet manipulation by JO-robot with magnetic particles aligned along the long side (JO-robot (0°)). (a) Schematic diagram of JO-robot (0°). The dotted arrow indicates the orientation of the carbonyl iron particles chains. The right part shows the carbonyl iron particles chains inside the JO-robot (0°). Scale bar is $100\ \mu\text{m}$. (b) Droplet manipulation by JO-robot (0°). Droplet can be dispensed by JO-robot (0°), but cannot be released. The insets show the relative position of the squeezed droplet and the JO-robot. The squeezed droplet is located on the side of the JO-robot. Scale bar is $2\ \text{mm}$.

When the angle between the carbonyl iron particles chains and the long side is set to 30° , the JO-robot (30°) folds along the creases and eventually tumbles around the short side as well (Figure R23). After contacting the water droplet, JO-robot (30°) can also achieve droplet dispensing by tumbling and wrapping part of the liquid. Similar to JO-robot (0°), **JO-robot (30°) is also difficult to release the dispensed daughter droplet due to the mismatch between the droplet location and the tumbling direction.**

Figure R23. Droplet manipulation by JO-robot with an angle of 30 degrees between the magnetic particles chains and the long side (JO-robot (30°)). (a) Schematic diagram of JO-robot (30°). The dotted arrow indicates the orientation of the carbonyl iron particles chains. The right part shows the carbonyl iron particles chains inside the JO-robot (30°). Scale bar is 100 μm . (b) Droplet manipulation by JO-robot (30°). Droplet can be dispensed by JO-robot (30°), but cannot be released. The insets show the relative position of the squeezed droplet and the JO-robot. The squeezed droplet is located on the side of the JO-robot. Scale bar is 2 mm.

When the angle between the carbonyl iron particles chains and the long side is set to 60 degrees, JO-robot (60°) tumbles around the long side. However, **the movement of JO-robot (60°) is very unstable**. The robot swings repeatedly during the movement. The unstable swing of the robot originates from the misalignment between the tumbling axis and the long side. The tumbling axis is perpendicular to the carbonyl iron particles chains, so the angle between the tumbling axis and the long side is 30 degrees. After wrapping the droplet, the JO-robot (60°) can also dispense the daughter droplet by tumbling. To release the droplet, the JO-robot (60°) turns the side carrying the droplet downward. When the magnetic field increases, the droplet between the robot and the superhydrophobic substrate can be squeezed out. After that, the JO-robot (60°) tumbles to detach the droplet for daughter droplet release (Figure R24). Although daughter droplet dispensing and release can be realized, **the controllability of the tumbling process and droplet manipulation process of the JO-robot (60°) is poor**.

Figure R24. Droplet manipulation by JO-robot with an angle of 60 degrees between the magnetic particles chains and the long side (JO-robot (60°)). (a) Schematic diagram of JO-robot (60°). The dotted arrow indicates the orientation of the carbonyl iron particles chains. The right part shows the carbonyl iron particles chains inside the JO-robot (60°). Scale bar is 100 μm . (b) Droplet manipulation by JO-robot (60°). Droplet can be dispensed and released by JO-robot (60°). Due to the misalignment between the tumbling axis and the long side, the controllability of the tumbling process and droplet manipulation process of the JO-robot (60°) is poor. Scale bar is 2 mm.

The comparison experiments show that JO-robot can realize stable tumbling motion and droplet dispensing and release functions only when the carbonyl iron particles chains are designed to be perpendicular to the long side (i.e., magnetic particles chains are aligned along the short side of the JO-robot) as demonstrated in the manuscript.

Revision: Following the reviewer's comments, we added the discussion on the effect of magnetic particles chains orientation on the tumbling motion and droplet manipulation functions of the JO-robot.

Page 5, Line 192 – Line 198: *When the chains are distributed perpendicularly to the long side of the robot, the tumbling axis overlaps with the long side. Therefore, the JO-robot can stably tumble around the long side to approach and manipulate the droplet under the action of the moving periodic magnetic field generated by magnets array (Supplementary Fig. 7, 8). In contrast, when the carbonyl iron particles are uniformly distributed, the robot cannot be driven by the magnetic field (Supplementary Fig. 9).*

Page 8, Line 309 – Line 312: *In addition, we further discuss the effects of different*

design parameters on droplet dispensing and release, such as the orientation of carbonyl iron particles chains, the number and distribution of creases, and the width and length of the JO-robot (Supplementary Fig.16-22).

Additional Figures (*Supplementary Figure 9, 16-18*) and *corresponding captions* have been added to the revised supplementary material.

6. In this work, gradual folding of the JO-exosuit is generated by the surface tension of the liquid, such as water, NaOH, and HCL. How long does it take to fold, and should the folding be gradual? Does the magnitude of the surface tension affect the folding time? Is it possible to transport the droplet with large surface tension (e.g., liquid metal), such as demonstration in S. Jhang et al., Adv. Mater. Interfaces, 10, 2201247 (2022)?

Response: Thanks for the reviewer's comment. When controlling the motion of the JO-robot using the moving magnetic field, the droplet is pulled by the robot and deformed. As discussed in the response to question #3, the droplet tends to reduce the liquid-air interface, thus lowering the surface energy at the cost of increasing the elastic potential energy of the elastic substrate, i.e., JO-robot folding. Therefore, the folding time is controlled by the moving of the magnets array (magnetic field). We tested the folding time at different magnetic field driving speeds. As the magnets array moving speed increases from 1 mm/s to 30 mm/s, the folding time of the JO-robot decreases from 14.48 ± 0.25 s to 0.52 ± 0.03 s (Figure R25). By further increasing the moving speed of the magnets array, the folding process of JO-robot can even be realized in 0.06875 s (Figure R26).

The influence of liquid surface tension on folding time is also tested experimentally. By mixing ethanol with water to prepare ethanol solutions with different volume ratios, liquids with different surface tensions can be obtained. We found that when the volume ratio of ethanol to water is 25:100, JO-robot can still effectively dispense and release the ethanol solutions. Therefore, we use ethanol solutions with ethanol volume ratios ranging from 0 (pure water) to 25% (volume ratio of ethanol to water is 25:100) for the test. The relationship between the liquid surface tension and the volume ratios of ethanol solution is shown in Figure R27a. When the volume ratio of ethanol to water increased from 0 to 25%, the liquid surface tension decreased rapidly from 72.7 mN/m to 42.4 mN/m. As can be seen in Figure R27b, the liquid surface tension has little effect on the folding time of the JO-robot. The folding time is between 1.42 s and 1.56 s when the moving speed of magnets array is 10 mm/s.

Figure R25. Dependence of the folding time of the JO-robot on the magnets array moving speed. As the magnets array moving speed increases from 1 mm/s to 30 mm/s, the folding time of the JO-robot decreases from 14.48 ± 0.25 s to 0.52 ± 0.03 s. The inset shows the entire folding process of the JO-robot.

Figure R26. The high-speed folding of JO-robot. The folding of the JO-robot can be realized in 68.75 ms. Scale bar is 1 mm. The folding process is captured by a high-speed camera (Phantom VEO 710 s, AMETEK, USA).

Figure R27. Influence of liquid surface tension on folding time of JO-robot. (a) Surface tension of ethanol solutions with different volume ratios. (b) Dependence of the folding time of the JO-robot on ethanol solutions with different volume ratios (magnets array moving speed: 10 mm/s).

Besides water droplet, liquids with high surface tension, such as liquid metal, can also be transported by JO-robot. The directional transport of liquid metal droplet has potential applications in the fields of soft electronics and microelectronics (*Adv. Mater. Interfaces* **10**, 2201247 (2022)). Here, we use the JO-robot to realize the controllable transportation of liquid metal (gallium-indium-tin alloys). As shown in Figure R28, the JO-robot actively approaches the liquid metal droplet by tumbling. The unmodified surface of the JO-robot comes into contact with the liquid metal and wraps the liquid metal under the action of the surface oxide layer. The laser-modified surface has rough micro/nanostructures, which gives it low adhesion to liquid metal (*ACS Appl. Mater. Interfaces* **12**, 55390-55398 (2020)) and cannot be adhered by liquid metal, while the unmodified surface has high adhesion to the liquid metal. Therefore, the Janus characteristic of JO-robot is still effective for liquid metal. After wrapping the liquid metal, JO-robot can carry the liquid metal on the microstructure surface for untethered transportation.

Figure R28. Liquid metal manipulation by the JO-robot. Scale bar is 2 mm.

Revision: Following the reviewer’s comments, the discussion on the folding time of the JO-robot and the transportation of liquid metal has been added to the revised manuscript and supplementary material (Supplementary Figure 2, 11, 12). The corresponding reference has been added as Ref. 28.

Page 4, Line 149 – Line 150: *Besides water droplet, liquids with high surface tension, such as liquid metal²⁸, can also be transported by JO-robot (Supplementary Fig. 2).*

In order to avoid misunderstandings and to describe the folding process more accurately, we have deleted “gradually” from the corresponding description.

Page 6, Line 228 – Line 235: *As the JO-robot tumbles, it folds along the creases by the surface tension of the liquid^{25,38,39} and wraps a part of the liquid. With further tumbling of the JO-robot, the tumbling axis converts to the edge II. Both two sides of the JO-robot fold under the surface tension until the cross-section becomes a closed triangle. The folding time is managed by the moving speed of the magnets array and is not affected by the liquid surface tension. The folding time can be controlled between ~68.75 ms to ~14.48 s (Supplementary Fig. 11, 12).*

Additional Figures (*Supplementary Fig. 2, 11, 12*) and *corresponding captions* have been added to the revised supplementary material.

7. **In the case of stirring and mixing droplets in Figure 4g-i, I think it is impossible to remove JO-exosuit from the droplets without further wrapping liquids. However, in Figure 5 and page 9 line 387, it seems to transfer JO-exosuit from a liquid to another without folding. Does it happen with certain types of liquid, like buffer solution? What does “sliding out JO-exosuit using magnetic attraction” mean? Please kindly explain this mechanism in detail.**

Response: We thank the reviewer for pointing this out. The JO-robot can remove from the droplet without further wrapping liquid (without folding). This process is not related to the type of liquid, but to the operation method.

As shown in Figure R29, we use a single neodymium-iron-boron (NdFeB) permanent magnet (20 mm × 20 mm × 10 mm) to remove the JO-robot without wrapping liquid (water). The magnetic field simulation is shown in Figure R29a. The N-pole and S-pole are distributed up and down along the magnet. The magnetic field strength at the edge of the magnet is larger than that in the middle. The sliding out process of JO-robot is shown in Figure R29b. First, the magnet approaches the JO-robot from the bottom. When approaching, the center of the magnet is located at the lower left of the JO-robot. JO-robot tilts to the right under the action of strong magnetic torque, and the plates on both sides of the robot unfold. Then the magnet moves to the right. The JO-robot gradually tumbles to the left along the magnetic field direction and squeezes out the liquid between it and the substrate. As the magnet keeps moving to the right, the angle between the JO-robot and the substrate gradually decreases until JO-robot is completely fit to the substrate. At this time, the liquid is squeezed back into the large droplet. As the magnet moves to the right, the JO-robot gradually approaches the edge of the magnet, where the magnetic attraction becomes larger. The JO-robot is dragged away from the droplet by magnetic attraction, thus enabling controlled removal without folding, i.e., sliding out JO-robot using magnetic attraction.

Figure R29. Removal of JO-robot without folding. (a) The magnetic field simulation. (b) Optical images of the detachment process.

Revision: Following the reviewer’s suggestion, we have added the discussion of this removal method to the revised manuscript and supplementary material (Supplementary Fig. 27 and Supplementary Movie 14).

Page 10, Line 399 – Line 401: *In addition, the JO-robot can be detached from the droplet in two ways after completing the manipulation targets (with and without folding, Supplementary Fig. 26, 27, Supplementary Movie 14).*

Page 11, Line 454 – Line 457: *(ii) The nucleic acid-bound JO-robot is carefully slid out along the water-oil interface using magnetic attraction (without folding, Supplementary Fig. 27) to minimize the carryover of the previous reagent, and then it is controlled to tumble towards the next reagent (washing buffer 1).*

Additional Figure (*Supplementary Fig. 27*), *corresponding caption* and revised *Supplementary Movie 14* have been added to the revised supplementary material.

8. Sentences in this manuscript need to be rearranged and revised to improve readability. There is lack of connection between contents and emphasis on key principles and novelties. For example, it would be better to move the sentence in page 6 line 244, “The Janus characteristics of the JO-exosuit ~” to the head of the section.

Response: We thank the reviewer for the comments. Sentences have been carefully revised and rearranged in the revised manuscript.

Revision: The sentence in page 6 line 244 “The Janus characteristics of the JO-exosuit ~” has been revised to “*During the dispensing and release processes, the Janus wetting characteristic of JO-robot plays the key role. The superhydrophobic outer surface is not wetted by droplet, which promotes the formation of the necking point, facilitates the dispensing of daughter droplet and avoids unnecessary liquid residue. It also ensures that the droplet does not stick to the JO-robot during release. Although other preparation methods such as 3D printing and ultraviolet lithography can easily prepare magnetic robots, they need additional surface modification to obtain Janus wetting characteristic^{29,30}. The comparison tests of the droplet dispensing and release of magnetic robot without the Janus characteristic are shown in Supplementary Fig. 13, 14.*”. It has been rearranged to the new position in the revised manuscript (page 6, line 255 to page 7, line 264).

Other similar sentences have also been carefully revised and rearranged in the revised manuscript.

9. page 3 line 102, “for directionally transport” should be changed to “to directionally transport” or “for directional transportation”.

Response: We thank the reviewer for the comment. The sentence has been revised.

Revision: Page 3, Line 105 – Line 107: *The JO-robot can tumble under the magnetic field and it can spontaneously wrap droplet for directional transportation.*

10. page 3 line 120, It would be better to name them “top and bottom side”, than “upper and lower side”, when describing two different sides with Janus wettability.

Response: We thank the reviewer for this suggestion. The “upper and lower side” has been renamed by “top and bottom side”.

11. page 5 line 177, Check the conjunctions between two sentences, “The modified surface is distributed with periodic bumped structures and covered with irregular micro-nanoparticles. While the unmodified surface is relatively flat.”.

Response: Thanks for pointing this out. The sentence has been revised.

Revision: Page 5, Line 209 – Line 211: *The modified surface is distributed with periodic bumped structures and covered with irregular micro-nanoparticles, while the unmodified surface is relatively flat.*

12. page 6 line 232, Check typo in “iterations”.

Response: Thanks for the reviewer's comment. Here we use the word “iterations” to mean that we use the same operation rule several times in the simulation to get the equilibrium shape of the droplet. We are still grateful to the reviewer for pointing this out.

13. page 8 line 325, There is no verb in sentence, “First, the JO-exosuit samples ~ larger droplet (~30 μ L), and then ~”.

Response: We thank the reviewer for pointing this out. The sentence has been revised.

Revision: Page 9, Line 388 – Page 10, Line 391: *First, the JO-robot dispenses a ~6.94 μ L sodium hydroxide (NaOH) droplet from the large droplet (~30 μ L), and then transports and merges it with a ~30 μ L hydrochloric acid (HCl) droplet.*

Finally, we sincerely appreciate the reviewer for these thoughtful and insightful comments. The manuscript has greatly benefited from these comments.

REVIEWER COMMENTS

Reviewer #1 (Remarks to the Author):

The authors addressed all issues in the revision.

The revised manuscript is improved well and is sufficient to be published. No more major comments, but proofreading is required.

Reviewer #2 (Remarks to the Author):

Remarks to the Author:

In the rebuttal letter and revised manuscript, the Authors have improved the demonstration and explanation to reveal the factors determining ability of droplet transportation, such as magnetic field conditions and design parameters of the JO robots. However, I recommend a few additions and revisions for valuable publication, as below.

1. As the author replied, the volume of the transportable droplet should be decreased for fast transportation under the fast-moving magnets array. Critical volume of the transportable droplet would also be determined by the design parameters and the number of the crease of the JO-robots. I believe that the author can organize and plot data of the critical volume depending on the several variables that have already been experimented.

2. The author simply indicated the applied magnetic flux density as average \pm standard deviation. However, as the reviewer denoted in Figure R29, a permanent magnet with N/S poles generates a magnetic field gradient which has different magnetic flux densities according to x-, y-, and z- directions. How about deconvoluting the directional magnetic flux density demanded for controlling droplets?

3. It is impressive that the tumbling axis perpendicular to the chain of the magnetic particles, and this arouses the failure and success of the droplet release. However, if the author summarizes the experimented data of the success/failure of dispensing/releasing droplets determined by the required time, the required magnetic flux density, and angle (0 to 90°) of magnetic particle chains in JO-robot as the 2D map (phase diagram) or table, it could be more concisely and clearly explained to

the readers. Examples of the 2D map can be seen in S. Won et al., *Nat. Commun.*, 13, 6750 (2022) and J. E. Park et al., *ACS Nano.*, 16, 3152 (2022).

4. Subsequent to previous question 7, the word "magnetic attraction" is still unkind and insufficient to understand how to JO-robot can be slid out without folding. Please revise sentences in the main manuscript.

Reply to the Reviewers' comments

Thank you very much for organizing reviews of our manuscript and we sincerely appreciate all the reviewers for their positive evaluations and valuable comments. The reviewer's comments have been carefully considered and some additional experiments have been carried out. The manuscript has been carefully revised to address the reviewers' concerns. The major revisions were highlighted **in red** in the revised manuscript. The point-to-point answers to the comments are listed below.

Reviewer #1 (Remarks to the Author):

The authors addressed all issues in the revision.

The revised manuscript is improved well and is sufficient to be published. No more major comments, but proofreading is required.

Response: We sincerely appreciate the reviewer for the professional and constructive comments. The overall quality of our study has greatly benefited during the last round of review.

Reviewer #2 (Remarks to the Author):

In the rebuttal letter and revised manuscript, the Authors have improved the demonstration and explanation to reveal the factors determining ability of droplet transportation, such as magnetic field conditions and design parameters of the JO robots. However, I recommend a few additions and revisions for valuable publication, as below.

1. As the author replied, the volume of the transportable droplet should be decreased for fast transportation under the fast-moving magnets array. Critical volume of the transportable droplet would also be determined by the design parameters and the number of the crease of the JO-robots. I believe that the author can organize and plot data of the critical volume depending on the several variables that have already been experimented.

Response: We thank the reviewer for the comment. As the reviewer pointed, maximum volume of the transportable droplet is determined by the design parameters of the JO-robots as well as the transporting speed. For the specified design parameters, there exists a critical transport volume at a particular transport speed. Similarly, a specific transport volume corresponds to a critical transport speed. Since the measurement of transport speed is considerably more accurate and straightforward compared to droplet volume, we dedicated significant time and effort to experimentally measure the critical transport speed of the droplets with a given volume, instead of the critical droplet volume at a given transport speed. Then we organized and plotted the relationship

curves between the droplet volume and transport speed under different JO-robot parameters.

Here we define the critical transport speed of the droplet as the maximum speed at which the droplet can be successfully transported without losses. When the transport speed is greater than the critical speed, some of the liquid can be lost during transportation. As shown in Figure R1, the droplet is thrown out of the JO-robot during transport and is retracted into the robot again as the robot tumbles. However, some of the liquid is "cut off" by the tumbling of JO-robot before the liquid is retracted, resulting in the loss of the liquid. Based on our experiment we can infer that the process of liquid loss is related to the volume of droplet and the transport speed of JO-robot. The larger the volume of the droplet, the easier it is for the droplet to be thrown out and cut off by the robot. The higher the speed of the JO-robot, the easier it is for the robot to cut off the liquid before retraction occurs.

Figure R1. Liquid loss during transportation by JO-robot (with three creases). The inset is the schematic diagram of JO-robot. Scale bar is 2 mm.

We have tested the relationship between droplet volume and critical transport speed. Figures R2-R5 show the transport process of droplet with varying volumes by the JO-robot with different number of creases (n), different crease distances (d), different widths (w) and different lengths (l) of the side plates, respectively. It is found that the critical droplet transport speed decreases obviously with the increase of droplet volume when manipulated by the JO-robot with the same parameters. However, in the experiments, we found that the critical transport speed is influenced by manifold and intricate factors.

For the JO-robots with identical parameters, the change in liquid volume affects the shape of JO-robots, and further results in different motion modes. The motion mode significantly influences the robot-droplet interaction and ultimately impacts the critical droplet transport speed. For the JO-robots with different parameters, even the droplets with the same volume can produce different shapes, which also affects the critical droplet transport speed.

For example, JO-robot (with the distance between the crease and the short side of 1 mm) forms a U-like shape when wrapping the droplet with the volume of 3.5 μL and tumbles

around the side of the U-shaped structure. When the droplet volume increases to 21.5 μL , the JO-robot can form an ellipsoidal shape when wrapping the droplet, and it tumbles on the droplet surface around the long side (Figure R3b). Similar phenomena occur when other JO-robots transport droplets of different volumes. When the droplet volume is small, the JO-robot can completely wrap the droplet and tumble around the side of the robot while transporting the droplet. When the droplet volume is large, the JO-robot cannot completely wrap the droplet and it tumbles along the droplet contour (Figures R2-R5). For another point of view, when JO-robots with different parameters wrap droplets with the same volume, it can be seen from Figures R2-R5 that the initial shapes of JO-robots are different, which also affects the critical transport speed. Moreover, as the droplet volume increases, the JO-robot exhibits sporadic twisting, bending or unfolding behaviors during transportation (Figures R2-R5), which also impacts the critical transport speed of droplets.

In conclusion, all the above mentioned factors make the interaction process between droplet and JO-robot complicated. At the same time, the effect of these influencing factors is not constant, meaning that as the parameters of the JO-robot or the droplet volume change, the degree to which these factors affect critical droplet transport speed also change. Hence the relationship between the critical droplet transport speed and the JO-robot parameters is not monotonic when the droplet volume is constant (Figure R6).

Figure R2. Droplet transport by JO-robots with different number of creases. (a) The schematic diagram of the JO-robots with different number of creases. (b-f) JO-robots with 0 to 4 creases are used to transport droplets of different volumes, respectively. Scale bar is 5 mm.

Figure R3. Droplet transport by JO-robots with different crease distributions. (a) The schematic diagram of the JO-robots with different crease distributions. (b-f) JO-robots with the distance between the crease and the short side from 1 mm to 3 mm are used to transport droplets of different volumes, respectively. Scale bar is 5 mm.

Figure R4. Droplet transport by JO-robots with different widths. (a) The schematic diagram of the JO-robots with different widths. (b-f) JO-robots with widths from 1 mm to 4 mm are used to transport droplets of different volumes, respectively. Scale bar is 5 mm.

Figure R5. Droplet transport by JO-robots with different side plate lengths. (a) The schematic diagram of the JO-robots with different side plate lengths. (b-f) JO-robots with side plate lengths from 1 mm to 4 mm are used to transport droplets of different volumes, respectively. Scale bar is 5 mm.

Figure R6. Experimental results of the critical transport speed versus droplet volume of JO-robots with different parameters. (a-d) Experimental results of the critical transport speed versus droplet volume of JO-robots with different number of creases (n), different crease distributions (the distance between the crease and the short side of the JO-robot is d), different widths (w) and different lengths of the side plate (l), respectively.

2. The author simply indicated the applied magnetic flux density as average \pm standard deviation. However, as the reviewer denoted in Figure R29, a permanent magnet with N/S poles generates a magnetic field gradient which has different magnetic flux densities according to x-, y-, and z-directions. How about deconvoluting the directional magnetic flux density demanded for controlling droplets?

Response: We thank the reviewer for the valuable comment. Figures R7-R11 show the magnetic flux density in the x-, y-, and z-directions of the workspace where the JO-robot is located during the tumbling, droplet dispensing and droplet transportation.

The processes including the tumbling of the JO-robot, the dispensing and transportation of droplet are manipulated by the horizontal movement of the magnets array. Because these motions (both the motion of the JO-robot and the motion of the magnets array) are a series of continuous processes, the whole process can be hardly described by a specific time and a specific position information (corresponding to a specific magnetic

flux density). Therefore, we measured the magnetic flux density information over the entire range of the magnets array in the working space of JO-robot (Figures R7-R11).

For the convenience of description, in the manuscript we simply used the average value of local maximum magnetic flux density in z-direction of the working space (at the position where the magnetic flux density is nearly zero in both the x- and y-directions) to describe the magnetic flux density information. The main droplet manipulation processes are carried out in the region away from both ends of the magnets array, so the statistical range of the magnetic flux density in calculating the average value is between the positions at 30 mm and 310 mm of the magnets array.

Here, we define successful droplet extrusion by the moment the droplet is squeezed out of the JO-robot. JO-robot with droplet is located above the junction of two magnets. With the magnets array approaching vertically from below, the droplet is eventually squeezed out of the JO-robot (Figure R12). At this point the magnetic flux density in the space where the JO-robot is located can give specific values. The magnetic flux densities in the x-, y-, and z-directions are measured to be 265.0 ± 24.7 mT, 0.5 ± 0.2 mT and 0.6 ± 0.2 mT, respectively.

Figure R7. The continuous tumbling process of JO-robot and the magnetic flux density information. (a) The continuous tumbling motion of the JO-robot (minimum driving magnetic field state). When the magnetic flux density decreases further, it leads to difficulties in tumbling motion of the JO-robot. Scale bar is 2 mm. (b) The magnetic flux density in the x-, y-, and z-directions of the workspace where the JO-robot is located during the continuous tumbling process (~21 mm away from the upper surface of the magnets array). The length in the horizontal coordinate refers to the length of the magnet array.

Figure R8. Continuous droplet dispensing process of the JO-robot (minimum driving magnetic field state) and the magnetic flux density information. (a) The continuous droplet dispensing process of the JO-robot (minimum driving magnetic field state). At this magnetic flux density, it is difficult for the JO-robot to dispense the droplet. Successful droplet dispensing is only possible when the magnetic flux density increases. Scale bar is 2 mm. (b) The magnetic flux density in the x-, y-, and z-directions of the workspace where the JO-robot is located during the continuous dispensing process (~12 mm away from the upper surface of the magnets array).

Figure R9. Continuous droplet dispensing process of the JO-robot (maximum driving magnetic field state) and the magnetic flux density information. (a) The continuous droplet dispensing process of the JO-robot (maximum driving magnetic field state). At this magnetic flux density, the plates on both sides of the JO-robot (the JO-robot can be seen as three plates connected by two creases) can be unfolded. A portion of the liquid

which adheres to the hydrophobic surface (with high droplet adhesion) is eventually dispensed. Although JO-robot is capable of dispensing droplet, it lacks the ability to transport the droplet. When JO-robot tumbles, it interacts with the substrate and squeezes out the droplet, resulting in the failure of transportation. Successful droplet dispensing by folding and wrapping, and subsequent transportation of droplet by the JO-robot is only possible when the magnetic flux density decreases. Scale bar is 2 mm. (b) The magnetic flux density in the x-, y-, and z-directions of the workspace where the JO-robot is located during the continuous dispensing process (~8 mm away from the upper surface of the magnets array).

Figure R10. Droplet transportation process by JO-robot (minimum driving magnetic field state) and the magnetic flux density information. (a) Droplet transportation process by JO-robot (minimum driving magnetic field state). At this magnetic flux density, it is difficult for the JO-robot to transport the droplet. Successful droplet transportation is only possible when the magnetic flux density increases. Scale bar is 2 mm. (b) The magnetic flux density in the x-, y-, and z-directions of the workspace where the JO-robot is located during the transportation (~16 mm away from the upper surface of the magnets array).

Figure R11. Droplet transportation process by JO-robot (maximum driving magnetic field state) and the magnetic flux density information. (a) Droplet transportation process by JO-robot (maximum driving magnetic field state). At this magnetic flux density, the droplet can be squeezed out by the robot during transportation, resulting in the failure of transportation. Successful droplet transportation is only possible when the magnetic flux density decreases. Scale bar is 2 mm. (b) The magnetic flux density in the x-, y-, and z-directions of the workspace where the JO-robot is located during the transportation (~ 5 mm away from the upper surface of the magnets array).

Figure R12. JO-robot with droplet is located above the junction of two magnets. As the magnets array approaches vertically from below, the droplet is eventually squeezed out of the JO-robot. The distance between the JO-robot and the upper surface of the magnets array is ~ 4 mm. Scale bar is 2 mm.

Information about the distance between the working plane of the JO-robot and the upper surface of the magnets array, and the magnetic flux density required for different droplet manipulation processes are shown in Table R1. A continuous droplet manipulation process usually consists of the tumbling motion of JO-robot, the dispensing and transportation of droplet, so the distance should be kept between 8 mm and 12 mm to

meet all the above manipulation requirements. When releasing the droplet, the distance between the magnets array and the working plane of JO-robot should be reduced (< 4 mm).

Table R1. Magnetic field information for different droplet manipulation functions.

Droplet manipulation function	Distance*	Required magnetic flux density**
Tumbling	<21 mm	>16.9±1.8 mT
Dispensing	8 mm~12 mm	81.5±5.7 mT~155.4±10.4 mT
Transportation	5 mm~16 mm	29.0±2.6 mT~227.4±13.9 mT
Releasing	<4 mm	>265.0±24.7 mT

* The distance between the working plane of the JO-robot and the upper surface of the magnets array.

** For the tumbling of JO-robot, the dispensing and transportation of water droplet, the required magnetic flux density is described by the average value of the local maximum magnetic flux density in z-direction of the working space of JO-robot. The required magnetic flux density for droplet releasing is described by the horizontal magnetic flux density of the working space of JO-robot.

According to the reviewer's comment, we conducted measurements of the magnetic flux density (x-, y- and z-directions) required for different droplet manipulation processes including the tumbling of the JO-robot, the dispensing and transportation of droplet (Figure R13). The corresponding figure and table have been added to the supplementary material (Supplementary Figure 24 and Table 3). In addition, we added the magnetic flux density information in the y- and z-directions when releasing the droplet. The revisions to the original manuscript are marked **in red**.

Page 6, Line 235 – Line 238: *In brief, on-demand daughter droplet release is achieved by deliberate squeezing and push-off motions (with the magnetic flux density larger than 265.0±24.7 mT, 0.5±0.2 mT and 0.6±0.2 mT in the x-, y-, and z-directions, respectively. Supplementary Movie 6).*

Page 7, Line 294 – Line 298: *The magnetic flux density required for droplet manipulation including the tumbling of JO-robot, the dispensing and transportation of droplet is between 81.5±5.7 mT and 155.4±10.4 mT (the magnetic field information is shown in Supplementary Fig. 24 and Supplementary Table 3).*

Figure R13. The magnetic flux density (x-, y- and z-directions) that enables the droplet manipulation functions including the tumbling of the JO-robot, the dispensing and transportation of droplet. (a) Minimum magnetic flux density for droplet manipulation. The corresponding distance between the JO-robot and the upper surface of the magnets array is ~ 12 mm. The inset shows the schematic of the magnets array. (b) Maximum magnetic flux density for droplet manipulation. The corresponding distance between the JO-robot and the upper surface of the magnets array is ~ 8 mm. The length in the horizontal coordinate refers to the length of the magnet array. For the convenience of description, the average value of the local maximum magnetic flux density in the z-direction of the working space of JO-robot (at the position where the magnetic flux density is nearly zero in both the x- and y- directions) is simply used to describe the magnetic flux density information. The main droplet manipulation processes are carried out in the region away from both ends of the magnets array, so the statistical range of the magnetic flux density in calculating the average value is between the positions at 30 mm and 310 mm of the magnets array. So, the magnetic flux densities in (a) and (b) are simply described by 81.5 ± 5.7 mT and 155.4 ± 10.4 mT, respectively.

3. It is impressive that the tumbling axis perpendicular to the chain of the magnetic particles, and this arouses the failure and success of the droplet release. However, if the author summarizes the experimented data of the success/failure of dispensing/releasing droplets determined by the required time, the required magnetic flux density, and angle (0 to 90°) of magnetic particle chains in JO-robot as the 2D map (phase diagram) or table, it could be more concisely and clearly explained to the readers. Examples of the 2D map can be seen in S. Won et al., *Nat. Commun.*, 13, 6750 (2022) and J. E. Park et al., *ACS Nano.*, 16, 3152 (2022).

Response: We thank the reviewer for this insightful suggestion. We summarized the success/failure states of various droplet manipulation functions by the JO-robots with different angles of magnetic particles chains by phase diagram (Figure R14). We also summarized the magnetic field information required for the successful manipulation of different droplet manipulation functions including the working distance of the JO-robot (the distance between the working plane and the upper surface of the magnets array) and the magnetic flux density using a table (Table R1). In the last round of review, we proved that the folding (dispensing) process of the JO-robot can be controlled from tens of milliseconds to more than ten seconds. Compared with the parameters of the JO-robot and the magnetic field, the effect of action time on the process of droplet dispensing/releasing is weak and not discussed here.

Figure R14. The droplet manipulation functions that can be realized by JO-robot with different carbonyl iron particles chains angles. The green hook means that the function can be realized. And the red cross means that the function cannot be realized.

Table R1. Magnetic field information for different droplet manipulation functions.

Droplet manipulation function	Distance*	Required magnetic flux density**
Tumbling	<21 mm	>16.9±1.8 mT
Dispensing	8 mm~12 mm	81.5±5.7 mT~155.4±10.4 mT
Transportation	5 mm~16 mm	29.0±2.6 mT~227.4±13.9 mT
Releasing	<4 mm	>265.0±24.7 mT

* The distance between the working plane of the JO-robot and the upper surface of the magnets array.

** For the tumbling of JO-robot, the dispensing and transportation of water droplet, the required magnetic flux density is described by the average value of the local maximum magnetic flux density in z-direction of the working space of JO-robot. The required magnetic flux density for droplet releasing is described by the horizontal magnetic flux density of the working space of JO-robot.

Following the reviewer's comment, we added the phase diagram and table to the revised supplementary material (Supplementary Figure 19 and Supplementary Table 3). In addition, the corresponding references have been added as Ref. 31 and 32.

4. Subsequent to previous question 7, the word "magnetic attraction" is still unkind and insufficient to understand how to JO-robot can be slid out without folding. Please revise sentences in the main manuscript.

Response: We thank the reviewer for the comment. The related sentences have been carefully revised in the revised manuscript.

Page 10, Line 427 – Page 11, Line 439: *The nucleic acid-bound JO-robot is carefully slid out along the water-oil interface without folding. Unlike the tumbling and droplet dispensing processes of the JO-robot controlled by the magnets array, the sliding out process is controlled by a single magnet and it is similar with the sliding out process in air (Supplementary Fig. 29). First, the JO-robot that suspended at the top of the droplet moves to the side of the droplet as the magnet approaches vertically from below. As the magnet gets closer, the JO-robot is subjected to a greater magnetic action than capillary action, so that the plates on both sides of the robot can overcome capillary folding to unfold. As the magnet moves to the right, the JO-robot tilts to the left. The angle between the JO-robot and the substrate decreases. As the magnet continues to move right, the JO-robot is finally attracted by the strong magnetic field and slides out along the contour of the droplet without folding. Then it is controlled to tumble towards the next reagent (washing buffer 1).*

Finally, we would like to express our sincere gratitude once again for the invaluable comment provided by the reviewer. We believe that the revisions greatly contribute to the improvement of our manuscript.

REVIEWERS' COMMENTS

Reviewer #2 (Remarks to the Author):

The revised manuscript and supporting information have been improved to describe various demonstrations of controlling droplets by capillary and magnetic forces. The authors made an effort to meet the recommendations for content, however, grammar check and proofreading are required.

Reply to the Reviewers' comments

Thank you very much for organizing reviews of our manuscript and we sincerely appreciate all the reviewers for their positive evaluations and valuable comments. The point-to-point answers to the comments are listed below.

Reviewer #2 (Remarks to the Author):

The revised manuscript and supporting information have been improved to describe various demonstrations of controlling droplets by capillary and magnetic forces. The authors made an effort to meet the recommendations for content, however, grammar check and proofreading are required.

Response: We sincerely appreciate the reviewer for the professional and constructive comments, which have helped us to improve the overall quality of our study. We also sought the assistance of a native English-speaking colleague to edit the grammar and refine the language.